# Unsupervised Meta-Learning via In-Context Learning

**Anna Vettoruzzo**[*]
Halmstad University, Sweden
anna.vettoruzzo@hh.se

**Lorenzo Braccaioli**[*]
University of Trento, Italy
lorenzo.braccaioli@unitn.it

**Joaquin Vanschoren**
Eindhoven University of Technology, Netherlands
j.vanschoren@tue.nl

**Marlena Nowaczyk**
Halmstad University, Sweden
marlena17nowaczyk@gmail.com

## Abstract

Unsupervised meta-learning aims to learn feature representations from unsupervised datasets that can transfer to downstream tasks with limited labeled data. In this paper, we propose a novel approach to unsupervised meta-learning that leverages the generalization abilities of in-context learning observed in transformer architectures. Our method reframes meta-learning as a sequence modeling problem, enabling the transformer encoder to learn task context from support images and utilize it to predict query images. At the core of our approach lies the creation of diverse tasks generated using a combination of data augmentations and a mixing strategy that challenges the model during training while fostering generalization to unseen tasks at test time. Experimental results on benchmark datasets showcase the superiority of our approach over existing unsupervised meta-learning baselines, establishing it as the new state-of-the-art. Remarkably, our method achieves competitive results with supervised and self-supervised approaches, underscoring its efficacy in leveraging generalization over memorization.

## 1 Introduction

Meta-learning, or *learning-to-learn*, enables models to accumulate knowledge from multiple tasks, allowing rapid adaptation and generalization to new tasks (Vettoruzzo et al., 2024; Vanschoren, 2019). Traditional meta-learning approaches typically rely on labeled data to construct tasks during meta-training. However, collecting large labeled datasets in real-world applications is challenging and often impractical. Unsupervised meta-learning (UML) methods address this issue by leveraging unlabeled data to learn transferable feature representations, enabling adaptation to new tasks with limited labeled data (Vettoruzzo et al., 2024).

Various approaches have been proposed to address the UML problem (Hsu et al., 2018; Jang et al., 2022; Khodadadeh et al., 2019; Kong et al., 2021; Lee et al., 2022; 2020). However, UML still faces several challenges. Existing UML methods often rely on simple data augmentations to construct the training tasks, while following the standard meta-learning task sampling pipeline for evaluation. This results in a significant difference between training and testing tasks, limiting generalization and often requiring fine-tuning on the test domain. Furthermore, existing UML approaches typically assume that the training and test datasets belong to the same domain. In our framework, we loosen this assumption resulting in a more challenging setting that necessitates a better model generalization compared to usual meta-learning applications. We refer to this as the *cross-domain* scenario.

In this paper, we propose a novel approach to UML that addresses these challenges by leveraging in-context learning within a transformer architecture (Dong et al., 2022; Min et al., 2022). In-context learning allows the model to use the context provided by a sequence of input-output pairs to make predictions on new input data. Inspired by recent advancements in large language models (LLMs) (Wei et al., 2022; Brown et al., 2020; Liu et al., 2022), we formulate meta-learning as a

---

[*]Equal contributions.

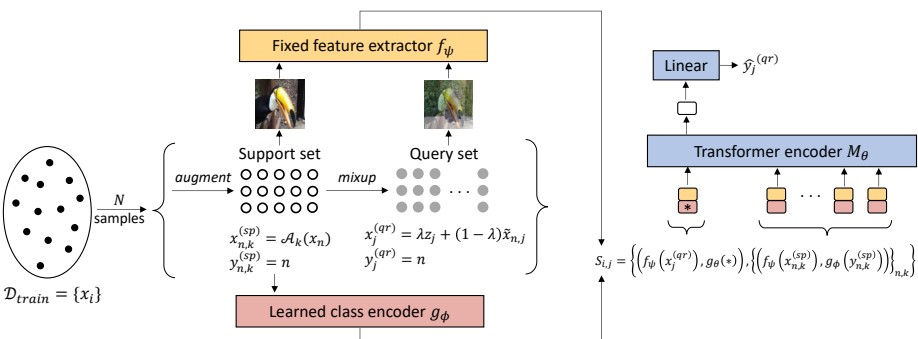

Figure 1: Visualization of CAMeLU (with 3-way 5-shot tasks). The left side illustrates the task creation mechanism, where $N$ samples are drawn from an unlabeled dataset $\mathcal{D}_{train}$. Each sample $x_n$ is augmented $K$ times to obtain $x_{n,k}^{(sp)}$. A strategy inspired by *mixup* (Zhang et al., 2018) is utilized for generating the query set by using an augmented version of $x_n$, i.e., $\tilde{x}_{n,j}$. The same pseudo-label $n \in [1, N]$ is assigned to all data generated from the sample $x_n$. On the right side, the so-created task is fed into the transformer encoder for predicting the query input. Inspired by CAML (Fifty et al., 2024), the transformer encoder processes demonstrations created by concatenating features from a fixed pre-trained feature extractor and a learned class encoder. The symbol $*$ denotes the unknown query label that the transformer encoder aims to predict.

sequence modeling problem, where a task is seen as a *non-causal* sequence of support images and an unknown query image. The support set is treated as the *context* utilized by the model to predict the class of the query image. We call our approach CAMeLU, which stands for **C**ontext-**A**ware **Me**ta-**L**earning in **U**nsupervised scenarios. Central to our approach is a novel task creation mechanism that enables the generation of a large number of different tasks from an unlabeled dataset. Drawing inspiration from the natural decision process of learning by analogy (Winston, 1980), we construct tasks that closely resemble the structure of those encountered during inference. Specifically, we use a combination of different data augmentation techniques based on basic image manipulations (Shorten & Khoshgoftaar, 2019) for generating the samples in the support set. Conversely, a strategy similar to *mixup* (Zhang et al., 2018) is employed to generate query images by combining a support element, after applying a distinct augmentation function, and an image randomly sampled from the training dataset. This process ensures that the query contains sufficient information from the support image to be classified as the latter, while introducing diversity by blending them. Consequently, query images appear distinct from their corresponding support images while still belonging to the same class, better mimicking the tasks seen at test time and hence enhancing generalization. Following task creation, support and query images are encoded using a fixed pre-trained feature extractor. The resulting latent representations are aggregated into a sequence and passed as input to a transformer encoder along with their label encodings. The transformer encoder learns to extract contextual information from support images and predict the query image in a single pass, eliminating the need for the fine-tuning step during inference. An overview of our approach is visualized in Fig. 1.

Throughout extensive experiments we demonstrate the effectiveness of the proposed approach to generalize to new tasks in real-time. Particularly, CAMeLU outperforms other UML baselines across several datasets, establishing itself as the state-of-the-art in the field. It also achieves comparable results to its supervised counterpart and to SSL approaches. While the latter requires fine-tuning on the test domain, CAMeLU obtains comparable performance with a single forward step, highlighting its applicability to real-time applications. Furthermore, by recasting the meta-learning phase as in-context learning within a transformer architecture, we improve efficiency, ensuring the whole training and inference phase can be executed with a consumer device with 8GB VRAM.

The main contributions of this paper are as follows:

- We introduce CAMeLU, a novel UML method that leverages in-context learning within a transformer architecture, reframing meta-learning as a sequence modeling problem.

- We propose a novel task creation mechanism that generates diverse few-shot tasks from unlabeled datasets using a combination of data augmentations and a mixing strategy. This

ensures better alignment between training and testing tasks, thus improving generalization performance.

- We demonstrate that CAMeLU outperforms existing UML baselines across five datasets, without the need for fine-tuning to the test domains.
- We investigate the ability of CAMeLU to generalize across various datasets, including those significantly different from the training data.

## 2 RELATED WORK

**Unsupervised meta-learning.** Meta-learning is a well-studied field in the machine learning community due to its ability to enable models to quickly adapt to tasks with limited labeled data. Pioneering work in the field (Finn et al., 2017; Snell et al., 2017; Vinyals et al., 2016; Mishra et al., 2018; Sung et al., 2018) considers the scenarios where a large labeled dataset is available for meta-training, a challenging requirement in real-world applications. UML addresses this challenge by extracting meaningful information from unsupervised data that can be transferred to downstream tasks with limited labeled data. Different techniques have been explored in the literature to construct diverse tasks. CACTUs (Hsu et al., 2018) applies clustering in the embedding space and assigns the same pseudo-label to all images in the same cluster. Other methods focus on generating synthetic samples, either using data augmentations, as in UMTRA (Khodadadeh et al., 2019), or leveraging interpolation in the latent space of a generative model (Khodadadeh et al., 2020). Differently, Meta-GMVAE (Lee et al., 2020) and Meta-SVEBM (Kong et al., 2021) use variational autoencoders and memory-based models for pseudo-label generation. Recent methodologies have also incorporated SSL techniques (Doersch et al., 2015) into UML methods. In particular, Set-SimCLR (Lee et al., 2022) builds on top of the SimCLR (Chen et al., 2020) approach and reframes meta-learning as a set-level problem, while PsCo (Jang et al., 2022), inspired by MoCo (He et al., 2020), utilizes a momentum encoder and a queue of previous samples to improve pseudo-labeling and construct diverse tasks for UML applications. Similarly, BECLR (Poulakakis-Daktylidis & Jamali-Rad, 2024) introduces an approach for unsupervised few-shot learning by proposing a constrastive representation learning framework, instead of meta-learning.

**Data augmentation.** Several UML approaches rely on data augmentation to construct the training tasks (Khodadadeh et al., 2019; Lee et al., 2022; Jang et al., 2022). However, traditional transformations such as rotation, translation, cropping, resizing, and flipping (Shorten & Khoshgoftaar, 2019) might generate images that are too similar to the original ones, ending up in tasks with low in-class variability between the support and query images. This creates a problem when the model needs to generalize to test tasks, where the query data are different instances than the support ones, not only augmented versions of them. In this paper, we addressed this limitation by generating query images using a strategy inspired by *mixup* (Zhang et al., 2018) to enhance model generalization. Similarly to *mixup*, which performs linear interpolation of the feature vectors at the pixel level, other strategies based on mixing images comprise CutMix (Yun et al., 2019), PatchMix (Liu et al., 2021), and Manifold Mixup (Verma et al., 2019).

**In-context learning.** In-context learning refers to the ability to perform a new task via inference alone by conditioning on a few input-output pairs and making predictions for new inputs (Dong et al., 2022). Although typical of LLMs (Devlin et al., 2019; Radford et al., 2019; Touvron et al., 2023), this ability has also been explored in different fields, such as in-painting (Bar et al., 2022; Zhang et al., 2024), image segmentation (Butoi et al., 2023), and notably meta-learning (Chan et al., 2022; Singh et al., 2024; Kirsch et al., 2022; Fifty et al., 2023; 2024; Min et al., 2022). Recent methods, such as Chan et al. (2022) and Singh et al. (2024), examine the emergence of in-context learning abilities from a data distribution perspective, extending these insights to images. GPICL (Kirsch et al., 2022) further demonstrates that transformers can be meta-trained as general-purpose in-context learners, while CAML (Fifty et al., 2024) adapts this concept to non-causal sequence modeling problems. Building on these advancements, our work takes a different direction by tackling the unsupervised meta-learning problem. Specifically, we introduce a novel task creation mechanism that, together with an in-context learner, enables learning directly from an unlabeled dataset. This approach differentiates our method from prior in-context learning techniques, aligning it with the unique requirements of UML.

# 3 PROPOSED APPROACH

Our proposed approach, *Context-Aware Meta-Learning in Unsupervised scenarios* (CAMeLU), leverages the in-context learning ability of transformers to address the challenges of UML. These challenges include the need to construct meaningful tasks from unlabeled data and the requirement for models to generalize effectively to new tasks during inference. CAMeLU consists of two phases that are intertwined during the model training. Initially, tasks are automatically constructed from an unlabeled dataset utilizing a combination of two strategies. Subsequently, we reformulate the meta-learning framework as a sequence modeling problem, aiming to harness the in-context learning capability of a transformer. This enables the model to extract context from the support samples and predict the unknown query samples without requiring any fine-tuning during the inference phase. The combination of these two phases is essential and guarantees good generalization performance without labeled information. Transformers excel at modeling dependencies and capturing relationships between support and query samples, which is particularly beneficial in few-shot learning scenarios. The novel task creation mechanism complements this by constructing diverse and challenging pseudo-tasks, effectively preparing the model for the complexities of target tasks. We delve into the two phases in Sect. 3.1 and Sect. 3.2, respectively.

## 3.1 TASK CREATION

Central to our proposed approach is the task creation mechanism. In meta-learning, a task $\mathcal{T}_i$ corresponds to a data generating distribution $\mathcal{T}_i \triangleq \{p_i(x), p_i(y|x)\}$, and consists of data from $N$ distinct classes. The data sampled from each task is divided into a *support set*, $D_i^{(sp)}$, containing $K$ training examples per class, and a *query set*, $D_i^{(qr)}$. At meta-test time, only the support set $D_{new}^{(sp)}$ of a task $\mathcal{T}_{new} \sim \mathcal{D}_{test}$ is labeled and used to fine-tune the model and make accurate predictions on the unlabeled query set. Contrary to supervised meta-learning, tasks in UML are only available at test time, while a large unlabeled dataset $\mathcal{D}_{train}$ is available during training. The main goal is to extract prior knowledge from this unlabeled dataset that can be generalized to a target task, $\mathcal{T}_{new} \sim \mathcal{D}_{test}$, during inference. A critical aspect of UML approaches lies in the task creation mechanism to create tasks from $\mathcal{D}_{train}$, which must ensure that the constructed training tasks reflect the structure of those encountered during testing, thereby facilitating effective generalization to novel tasks at test time. To do so, we employ two distinct strategies for constructing the support and query sets of each task.

For the support set, we randomly sample $N$ images from $\mathcal{D}_{train}$ under the assumption that they belong to distinct categories, as shown in Fig. 1. This assumption is reasonable when $N << C$, where $C$ denotes the total number of classes in $\mathcal{D}_{train}$, which is satisfied using a large training dataset. If we assume that all samples are equally distributed among the classes, i.e., $m$ samples per class, the probability that two or more samples are in the same class is equal to

$$\mathrm{P} = 1 - \frac{(C \cdot m) \cdot ((C-1) \cdot m) \cdots ((C-N+1) \cdot m)}{(C \cdot m) \cdot (C \cdot m - 1) \cdots (C \cdot m - N + 1)} = 1 - \frac{C! \cdot m^N \cdot (C \cdot m - N)!}{(C-N)! \cdot (C \cdot m)!}.$$

For example, the probability for a 5-way classification on the ImageNet-964 dataset used in our experiment is around $0.01$, which is negligible. To emulate the $K$-shot scenario typical of meta-learning tasks, we augment each of the $N$ images $K$ times, with an augmentation function $\mathcal{A}_k$ sampled from a predefined set of transformations $\mathcal{A}$, and we assign the same pseudo-label $n \in [1, N]$ to all data generated from the same sample $x_n$. Specifically, for each image $x_n$, $K$ augmentation functions are applied to obtain $x_{n,k}^{(sp)} = \mathcal{A}_k(x_n)$ with $\mathcal{A}_k \sim \mathcal{A}$ and $k = 1, \ldots, K$. One requirement of $\mathcal{A}_k$ is that the function must preserve class membership, i.e., $x_n \in c \rightarrow \mathcal{A}_k(x_n) \in c$, for $c \in C$. Although this property cannot be directly verified due to the lack of class information in the training set, it is reasonable to assume that it holds by selecting transformations that minimally alter the image content.

For the query set, we employ a different approach. We demonstrate in Appendix A.5 that simply applying data augmentations sampled from $\mathcal{A}$ is not sufficient for creating a query set resembling those in test tasks. At test time, the query set samples are different instances belonging to the same $N$ classes encountered in the support set, not augmented versions of the support samples. However, since $\mathcal{D}_{train}$ is unlabeled, we need a strategy to create new samples with the same implicit classes as those in the support set. For each query image $x_j^{(qr)}$ that we want to generate, we randomly select

an image $x_n$ from the ones sampled for the support set generation and we apply an augmentation function $\mathcal{A}_j \sim \mathcal{A}$, possibly different from the one used for the support generation. We then propose a new strategy inspired by *mixup*, where we combined the augmented image $\tilde{x}_{n,j} = \mathcal{A}_j(x_n)$ and an image $z_j$ sampled from $\mathcal{D}_{train}$ according to:

$$x_j^{(qr)} = \lambda z_j + (1 - \lambda)\tilde{x}_{n,j} \tag{1}$$

where $\lambda \sim Beta(\alpha, \beta)$ with $\alpha = 1, \beta = 1$ and $\lambda \in (0, 0.5)$, and $x_j^{(qr)}$ is assigned the same label $n$ as the support samples generated from $x_n$. By merging a small proportion of a new image $z_j$ into $\tilde{x}_{n,j}$, we enhance diversity in the query set with respect to the images in the support set. This strategy enforces the model to extract robust features and effectively generalize to scenarios where query images differ from the support samples, as commonly encountered at test time.

This task-creation mechanism can be seen as a task augmentation strategy (Yao et al., 2021; Rajendran et al., 2020) that allows the generation of a large number (almost infinite) of diverse tasks. This is particularly useful for meta-learning and in-context learning applications where the model needs to acquire knowledge from a multitude of tasks to generalize to unseen tasks sampled from different domains.

**Differences with *mixup*.** While the strategy used for generating the query images draws inspiration from the *mixup* strategy proposed in Zhang et al. (2018), there are some substantial differences. The aim of *mixup* is to develop a new data augmentation strategy to expand the number of training examples and diversify the data distribution used for training, thereby enhancing the robustness and generalization of neural networks. In CAMeLU, the primary objective of merging images is to encourage the model to learn even in scenarios where only a fraction of the class context is present in the image. In CAMeLU, $\lambda$ is sampled from a uniform distribution (obtained with a $Beta$ distribution with $\alpha = 1, \beta = 1$) in $(0, 0.5)$, guaranteeing that the amount of information from $z_j$ that is embedded into $x_j^{(qr)}$ is less than 50%, thus ensuring that the assigned label is consistent with the class of the support images generated from $x_n$. Indeed, we assign the same label $n$ to $x_j^{(qr)}$, forcing the network to learn to retrieve information in $x_j^{(qr)}$ that is related to the category of $x_n$. Contrarily, *mixup* creates new examples by interpolating both images and labels at the scope of limiting memorization over the training distribution.

## 3.2 IN-CONTEXT LEARNING METHOD

Following task creation, we rephrase the meta-learning framework as a non-causal sequence modeling problem, where the order of the examples does not entail a causal relationship. Inspired by recent developments in LLMs (Garg et al., 2022; Li et al., 2023; Devlin et al., 2019; Radford et al., 2019; Touvron et al., 2023), we treat each task as a prompt, where the support embeddings, together with the learned projected labels, form the demonstration context, whereas the query represents the classification problem that the network is required to solve. A model is said to *in-context learn* a task if it can approximate $y_j^{(qr)}$ for a new query input $x_j^{(qr)}$ by conditioning on a sequence $S_{i,j}$ containing in-context (support) examples and one query input defined as follows:

$$S_{i,j} = \left( (x_1^{(sp)}, y_1^{(sp)}), \ldots, (x_{NK}^{(sp)}, y_{NK}^{(sp)}), x_j^{(qr)} \right), \quad j = 1, \ldots, Q, \tag{2}$$

with $Q$ the number of query samples to classify and $NK$ the total number of context (support) samples. Formally, $M_\theta$ can in-context learn a task $\mathcal{T}_i$ if it can predict $y_j^{(qr)}$ with an average error

$$\mathbb{E}\left[ \sum_{j=1}^{Q} \ell(M_\theta(S_{i,j}), y_j^{(qr)}) \right] < \epsilon, \tag{3}$$

where $\ell$ is the loss function, $S_{i,j}$ is the sequence associated to $x_j^{(qr)}$ in $\mathcal{T}_i$, and $y_j^{(qr)} \in [1, N]$.

To achieve this, we design a model comprising three components: (1) a feature extractor $f_\psi$, (2) a class encoder $g_\phi$, and (3) a transformer encoder with a linear projection layer on top, i.e., $M_\theta$. The feature extractor aims to map support and query samples into a latent space where images with similar characteristics and semantic meaning are assigned similar representations. In

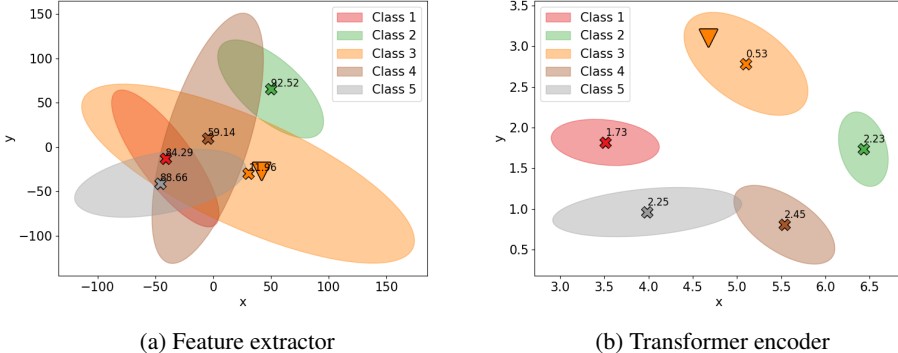

(a) Feature extractor  (b) Transformer encoder

Figure 2: Visualization of clustered embeddings obtained with CAMeLU after the feature extractor (left) and the transformer encoder (right) on a 5-way 5-shot task sampled from the CUB dataset. Crosses indicate the centroids of each class, and the numbers denote the Euclidean distances between the query (triangle) and each class centroid. The plots are obtained using t-SNE (Van der Maaten & Hinton, 2008) with a perplexity equal to 9.

Appendix A.4 we explore various feature extractors for this purpose, including those pre-trained via a supervised approach or leveraging an SSL technique. The resulting representations are then concatenated with a class embedding. The class embeddings for the support representations are generated by encoding the corresponding classes using the class encoder $g_\phi$. However, as the classes of the queries are unknown, a randomly initialized learnable vector is appended to each query representation. The so-combined embeddings are then organized into sequences $S_{i,j} = \left( (f_\psi(x_1^{(sp)}), g_\phi(y_1^{(sp)})), \ldots, (f_\psi(x_{NK}^{(sp)}), g_\phi(y_{NK}^{(sp)})), f_\psi(x_j^{(qr)}) \right), j = 1, \ldots, Q$, resembling the one in Eq. 2. These sequences are fed into the transformer encoder, and only the transformer output corresponding to the query sample is selected and passed through a projection layer to predict the query label. This process iterates for all queries in the task, and the aggregated loss is employed for model training. In particular, the training process can be formulated as an optimization process where the objective is as follows:

$$\min_{\theta, \phi} \; \mathbb{E}_{S_i} \left[ \frac{1}{Q} \sum_{j=1}^{Q} \ell(M_\theta(S_{i,j}), y_j^{(qr)}) \right] \tag{4}$$

with $S_i = \{S_{i,j}\}_{j=1}^Q$ denoting the set of sequences associated to each task $\mathcal{T}_i$ generated from $\mathcal{D}_{train}$ and $\ell$ is the cross-entropy loss function.

During evaluation, when a new task is presented, the available examples in $D_{new}^{(sp)}$ are utilized as contextual information to guide the classification of the query samples without requiring any fine-tuning or adaptation steps.

**Analysis.** To gain a better understanding of how the in-context learner functions during inference, we show the embedding space of an exemplary test task after the feature extractor and the transformer encoder in Fig. 2. The embedding space after the feature extractor appears sparse, with a large Euclidean distance between the query sample and the centroid of each class, which indicates limited class separability and less informative representations. In contrast, the transformer encoder significantly improves the representation, producing more compact and well-separated clusters. Notably, the query representation aligns closely with the support examples of the same class, showcasing the effectiveness of the transformer in utilizing the task context to refine predictions. This demonstrates the in-context learner's ability to adapt representations dynamically based on the task context and also provides evidence supporting CAMeLU's superior performance, particularly in cross-domain and few-shot scenarios where generalization is more challenging. Experiments for the other datasets can be found in Appendix A.2.

## 4 EXPERIMENTS

In this section, we demonstrate the effectiveness of CAMeLU across different datasets, and we compare the results with several baseline methods. In particular, we provide a quantitative comparison with UML baselines in Sect. 4.3, and we highlight the ability of CAMeLU to leverage generalization over memorization in Sect. 4.4. We then present the results with a small-scale training dataset in Sect. 4.5 and a comparison with SSL methods in Sect. 4.6.

### 4.1 DATASETS AND BASELINES

For the evaluation, we use two generic object recognition datasets, i.e., *mini*ImageNet (Ravi & Larochelle, 2016) and CIFAR-fs (Bertinetto et al., 2019), and three fine-grained image classification datasets, i.e., CUB (Wah et al., 2011), Aircraft (Maji et al., 2013), and Meta-iNat (Wertheimer & Hariharan, 2019). While *mini*ImageNet and CIFAR-fs share some classes with ImageNet-1k, CUB, Aircraft, and Meta-iNat focus on more specialized domains, ensuring a rigorous cross-domain evaluation. Each dataset is split into training, validation, and test sets following the splits in Ravi & Larochelle (2016) and Bertinetto et al. (2019) for *mini*ImageNet and CIFAR-fs, respectively, and in Triantafillou et al. (2019) and Poulakakis-Daktylidis & Jamali-Rad (2024) for the remaining datasets. All labels are removed from the datasets during the training phase.

We compare CAMeLU with standard UML approaches such as CACTUs (Hsu et al., 2018), UMTRA (Khodadadeh et al., 2019), Meta-GMVAE (Lee et al., 2020), and PsCo (Jang et al., 2022). These methods are evaluated in-domain as recommended in the original papers, with training and testing performed on the same dataset. While this setup is relatively simpler than the cross-domain evaluation employed for CAMeLU, applying these methods in a cross-domain scenario may not be fair, as they were not explicitly designed for such a challenging scenario. Only PsCo (Jang et al., 2022) is further evaluated in a cross-domain setting, as the authors demonstrate its adaptability to this scenario through an additional adaptation phase to the test domain. We also compare CAMeLU with BECLR (Poulakakis-Daktylidis & Jamali-Rad, 2024) in Sect. 4.5, a contrastive framework for unsupervised few-shot learning, and CAML (Fifty et al., 2024), a supervised meta-learning approach that assumes tasks are available both during the training and testing phases and leverages the in-context ability of transformer architectures to generalize to new tasks. Due to their similarities, we refer to CAML as the supervised counterpart of CAMeLU. Furthermore, Sect. 4.6 provides a comparative analysis with two fine-tuned state-of-the-art self-supervised trained networks, namely SimCLR (Chen et al., 2020) and SwAV (Caron et al., 2020).

### 4.2 TRAINING DETAILS

We report the results following the $N$-way $K$-shot classification task typical of meta-learning algorithms, where $N = 5$ and $K = 1$ or $K = 5$. All models are trained for $100$ epochs with $500$ episodes per epoch. Fine-tuning at test time ($100$ steps) is applied only if required. For CAMeLU, we do not apply any fine-tuning step to demonstrate the strength of its training stage, which does not require additional parameter updates during inference. Furthermore, we introduce ImageNet-964, a variant of ImageNet-1k (Deng et al., 2009) where classes from the validation and test splits of *mini*ImageNet are removed to prevent data leakage—a problem that is not taken into consideration by previous studies (Fifty et al., 2024; Jang et al., 2022). To provide a fair comparison, all cross-domain methods are trained on ImageNet-964. For CAMeLU, we use a ResNet-50 (He et al., 2016) feature extractor pre-trained on ImageNet-964 and a class encoder that maps one-hot label vectors to a 256-dimensional space. In Appendix A.4, we also report the results with different feature extractors. The transformer encoder consists of 8 layers, each with an eight-head self-attention block, an MLP, and a single projection layer that maps the transformer output to the predicted category. The model is trained with the Adam optimizer with a learning rate of $10^{-5}$ and a warmup cosine scheduler (Vaswani et al., 2017). To account for statistical variations, each algorithm is run three times in full, and the complete results reporting the standard deviations are presented in Appendix A.10. The experiments are executed using Python and the PyTorch library on an Nvidia GeForce RTX 3070 Ti Laptop GPU with 8GB of VRAM, while ablation studies and competitors are executed on an Nvidia A100-SXM4 GPU with 40GB of VRAM. More details about the training settings can be found in Appendix A.1, and the code is available at `https://github.com/bracca95/CAMeLU.git`.

Table 1: Performance comparison on *mini*ImageNet, CIFAR-fs, CUB, Aircraft, and Meta-iNat datasets for 5-way 1-shot and 5-way 5-shot scenarios. Cross-domain approaches are trained using ImageNet-964 and a ResNet-50 feature extractor. The symbol † indicates results that are affected by data leakage. The bold font highlights the best performing UML approach for each setting. Results show the average across three complete runs of the algorithms. Complete results with standard deviations are reported in Tab. 12 in Appendix A.10.

| Method | *mini*ImageNet 5w1s | *mini*ImageNet 5w5s | CIFAR-fs 5w1s | CIFAR-fs 5w5s | CUB 5w1s | CUB 5w5s | Aircraft 5w1s | Aircraft 5w5s | Meta-iNat 5w1s | Meta-iNat 5w5s |
|---|---|---|---|---|---|---|---|---|---|---|
| **In-Domain** | | | | | | | | | | |
| CACTUs-MAML | 43.30 | 54.21 | 42.00 | 56.64 | 31.19 | 36.81 | 24.06 | 27.26 | 20.13 | 21.84 |
| CACTUs-ProtoNet | 48.85 | 62.52 | 50.90 | 64.52 | 33.93 | 44.41 | 26.27 | 30.88 | 27.30 | 29.08 |
| UMTRA | 39.93 | 50.73 | 32.93 | 46.13 | 27.06 | 36.6 | 22.40 | 31.73 | 28.96 | 37.12 |
| Meta-GMVAE | 55.38† | 65.10† | 52.02 | 64.18 | 33.59 | 39.09 | 24.83 | 27.60 | 34.22 | 40.23 |
| PsCo | 47.29 | 64.85 | 42.21 | 62.92 | 33.09 | 51.02 | 26.19 | 38.80 | 36.97 | 55.88 |
| **Cross-Domain** | | | | | | | | | | |
| PsCo | 67.89 | 90.17 | 53.34 | 76.22 | 43.35 | 70.19 | 29.87 | 38.20 | 46.21 | 70.05 |
| **CAMeLU** | **76.51** | **92.14** | **61.79** | **80.43** | **65.52** | **80.35** | **33.17** | **39.11** | **57.27** | **75.45** |
| CAML (supervised) | 81.75 | 92.31 | 59.44 | 75.27 | 54.63 | 66.81 | 28.92 | 32.06 | 50.86 | 67.07 |

## 4.3 COMPARATIVE RESULTS

Table 1 provides an overview of the experimental results for both the 5-way 1-shot and the 5-way 5-shot scenarios. The results demonstrate that CAMeLU outperforms the existing UML methods, regardless of the difference in the evaluation setting. As highlighted in Sect. 4.1, CACTUs, UM-TRA, and Meta-GMVAE are evaluated only in-domain, requiring knowledge about the test domain prior to training. This is not necessary for CAMeLU as it demonstrates high performance in the challenging cross-domain scenario. Even compared to PsCo, the only UML method designed for cross-domain applications, CAMeLU exhibits a performance improvement across all datasets. Furthermore, PsCo requires a fine-tuning phase to adapt to the test domain, whereas CAMeLU achieves good performance with a single forward pass, enhancing its applicability to real-time applications. It is also worth noting that CAMeLU achieves comparable performance to its supervised counterpart, CAML, when evaluated on *mini*ImageNet and it even outperforms CAML when evaluated on more dissimilar domains, such as CUB, Aircraft, and Meta-iNat. This finding highlights the efficacy of the task construction strategy used in CAMeLU, which acts as a sort of task augmentation and enhances the generalization capability of the model.

## 4.4 MEMORIZATION TO GENERALIZATION PHASE SHIFT

During the training of CAMeLU, we observed a distinct trend in the validation accuracy, similar to the findings in Kirsch et al. (2022). Fig. 3 illustrates this pattern, showing the validation accuracy relative to its initial value, or, in other words, how much the model learns from datasets different from the one we are training on. Specifically, the curves in Fig. 3 resemble a logistic curve, which can be divided into three phases that we denote as *memorization*, *learning*, and *generalization*. In the memorization phase, the model memorizes the tasks seen during training and extends this knowledge to unseen tasks, resulting in a slight improvement for datasets with high similarity with ImageNet-964 (e.g., *mini*ImageNet and CIFAR-fs). For the other datasets, instead, transferring this knowledge can even result in a performance decrease due to the intrinsic domain distance of the dataset (see CUB and Aircraft,

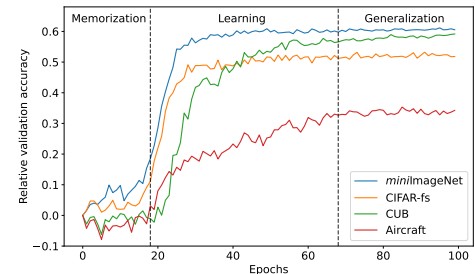

Figure 3: Analysis of learning behavior when transferring knowledge from a different prior dataset. The relative validation accuracy shows the difference between the current and first epoch accuracy on the validation set of *mini*ImageNet, CIFAR-fs, CUB, and Aircraft. CAMeLU is trained with ImageNet-964.

Table 2: Accuracy results obtained training PsCo, BECLR, and CAMeLU with a small-scale dataset, namely *mini*ImageNet, denoted as (*mini*) in the table. Results show both in-domain performance (on the test set of *mini*ImageNet) and cross-domain performance on CIFAR-fs, CUB, Aircraft, and Meta-iNat. The average results across three complete runs of the algorithms are reported. Complete results with standard deviations are presented in Tab. 13 in Appendix A.10.

|  | *mini*ImageNet | | CIFAR-fs | | CUB | | Aircraft | | Meta-iNat | |
|---|---|---|---|---|---|---|---|---|---|---|
|  | 5w1s | 5w5s | 5w1s | 5w5s | 5w1s | 5w5s | 5w1s | 5w5s | 5w1s | 5w5s |
| PsCo (*mini*) | 47.29 | 64.85 | 42.21 | 62.92 | 33.09 | 51.02 | 26.19 | **38.80** | 36.97 | 55.88 |
| BECLR (*mini*) | **81.04** | 87.88 | 57.05 | 72.82 | 42.47 | 58.03 | 27.48 | 38.46 | 49.87 | 65.05 |
| CAMeLU (*mini*) | 75.99 | **90.38** | **61.25** | **78.79** | **60.60** | **74.77** | **31.39** | 36.52 | **55.60** | **72.12** |

which are fine-grained datasets). As training progresses and the model observes more tasks, the learning phase occurs. This phase is characterized by a transition to the learning-to-learn state where the model learns to identify the tasks and to extract the features that are more useful for solving them. The duration of this phase varies, with datasets like *mini*ImageNet and CIFAR-fs exhibiting rapid learning within approximately 10 epochs, while datasets such as CUB and Aircraft may necessitate up to 40 epochs. This timespan depends on several factors, including the similarity between the training and evaluation datasets, the size of the test dataset, and the model's learning ability (Kirsch et al., 2022; Power et al., 2022). For instance, CUB, with its fine-grained nature and small test set size (around 1770 images), necessitates a longer learning phase compared to the *mini*ImageNet dataset (which has a test set with 12 000 images). Subsequently, in the generalization phase, the model can generalize to tasks significantly different from those observed during training using a single forward pass. Further analyses about the generalization capabilities of CAMeLU and the number of epochs required for reaching the generalization phase are presented in Sect. 4.5 and Appendix A.8.

## 4.5    GENERALIZATION ON SMALL-SCALE DATASETS

While most studies on training transformer architectures focus on large-scale training datasets, we investigate the generalization capabilities of CAMeLU using a small-scale training dataset. Specifically, we train CAMeLU on *mini*ImageNet and evaluate its performance both in-domain (i.e., on the test set of *mini*ImageNet) and cross-domain on CIFAR-fs, CUB, Aircraft, and Meta-iNat. CAMeLU demonstrates effective generalization in this scenario, showing impressive performance in both in-domain and cross-domain settings, as shown in Tab. 2, surpassing PsCo and BECLR by a large margin.

Additionally, a comparison of CAMeLU's performance when trained on a small-scale dataset like *mini*ImageNet (Tab. 2), on ImageNet-964 (Tab. 1), and on a large-scale dataset (Tab. 5 in Appendix A.3) show that our method is only slightly affected by the size of the training dataset. This robustness enhances CAMeLU's applicability to scenarios where only a small unlabeled training dataset is available, which is common in real-world applications.

Finally, Fig. 4 shows the relative validation accuracy of CAMeLU and CAML while trained and evaluated on *mini*ImageNet. While the curve obtained with CAMeLU reflects the three phases of *memorization*, *learning*, and *generalization* discussed in Sect. 4.4, the relative validation accuracy of CAML remains flat. This difference may be attributed to the task creation mechanism of CAMeLU, which acts as a task augmentation strategy, increasing the variability of tasks presented to the model during training and thereby enhancing its generalization capabilities.

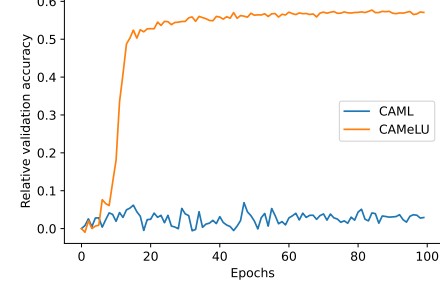

Figure 4: Relative validation accuracy of CAMeLU (orange) and CAML (blue) when evaluated in-domain on *mini*Imagenet and computed as in Sect. 4.4. The curve obtained with CAMeLU reflects the three phases of *memorization*, *learning*, and *generalization* even when using a small-scale dataset.

Table 3: Comparison between CAMeLU and SSL approaches for the 5-way 1-shot and 5-way 5-shot scenario on *mini*ImageNet, CIFAR-fs, CUB, Aircraft, and Meta-iNat. The symbol † indicates results that are affected by data leakage. Results show the average across three complete runs of the algorithms. Complete results with standard deviations are reported in Tab. 14 in Appendix A.10.

| Method | *mini*ImageNet | | CIFAR-fs | | CUB | | Aircraft | | Meta-iNat | |
|---|---|---|---|---|---|---|---|---|---|---|
| | 5w1s | 5w5s | 5w1s | 5w5s | 5w1s | 5w5s | 5w1s | 5w5s | 5w1s | 5w5s |
| SimCLR | **83.32**† | 94.86† | 64.52 | 84.36 | 47.35 | 66.87 | 29.36 | 39.99 | 52.44 | 73.19 |
| SwAV | 74.83† | **94.96**† | **66.97** | **87.14** | 47.84 | 69.31 | 30.33 | **47.43** | 53.57 | 74.53 |
| **CAMeLU** | 76.51 | 92.14 | 61.79 | 80.43 | **65.52** | **80.35** | **33.17** | 39.11 | **57.27** | **75.45** |

## 4.6 COMPARISON WITH SSL METHODS

In this section, we compare CAMeLU with SimCLR (Chen et al., 2020) and SwAV (Caron et al., 2020). For training SSL methods in our experiments, we employed a backbone network with a ResNet-50 architecture pre-trained on ImageNet-1k and obtained from PyTorch Lightning Bolts (Borovec et al., 2022). While this setup leads to data leakage when evaluated on *mini*ImageNet, due to overlap between the test and training classes, pre-training these SSL approaches from scratch using a different training dataset was beyond our available computational resources. To facilitate model adaptation to the test domain, we fine-tuned a classification layer on top of the pre-trained backbone using SGD with an initial learning rate of $0.01$, momentum of $0.9$, weight decay of $10^{-4}$, and 100 fine-tuning steps per task, following Jang et al. (2022). This setup differs from the evaluation setting used in CAMeLU, where predictions are obtained with a single forward pass, leveraging the in-context learning ability of transformer architectures. However, SSL approaches must adapt to the test domain before label predictions, resulting in a less challenging evaluation setting than CAMeLU. Results for SSL approaches are averaged over 500 test tasks and presented in Tab. 3. While SSL approaches outperform CAMeLU on *mini*ImageNet and CIFAR-fs, their performance decreases when evaluated on the other datasets. CUB, Aircraft, and Meta-iNat are fine-grained datasets significantly different from ImageNet-1k, challenging the transferability of features learned by SSL methods to these datasets. Moreover, the high performance on *mini*ImageNet and CIFAR-fs may be attributed to the presence of data leakage with ImageNet-1k and the high similarity with CIFAR-fs, as discussed in Sect. 4.2. CAMeLU, in contrast, demonstrates effective generalization to tasks sampled from these datasets, once again highlighting its generalization ability over mere memorization.

## 5 CONCLUSION

In this paper, we introduce CAMeLU, a novel approach for UML that leverages the in-context learning capabilities of transformer architectures to extract context from the support samples and make effective predictions on the query data. CAMeLU reframes meta-learning as a sequence modeling problem, where support images provide task context for predicting query images. At the core of CAMeLU is a novel task creation mechanism that generates diverse tasks from an unlabeled dataset, promoting effective generalization to unseen tasks. Our experimental results showcase the superiority of CAMeLU over existing UML methods, highlighting the applicability of the proposed method to domains different from the training one. Notably, CAMeLU can generalize to new domains with a single forward pass (real-time predictions), and it even outperforms its supervised counterpart thanks to its task creation mechanism. Furthermore, the proposed model can be stored and trained with a single GPU with only 8GB of VRAM, underscoring its efficiency in learning-to-learn in-context, rather than using a meta-training phase typical of previous meta-learning approaches.

Future research directions may explore extensions of CAMeLU to more complex domains, as well as investigations into further improving the task creation mechanism for enhanced generalization. It would be interesting to incorporate SSL techniques to obtain more robust feature representations and enhance generalization capabilities. Additionally, conducting further investigation into CAMeLU's ability to encourage generalization over memorization would provide valuable insights into its learning dynamics and potential areas for improvement.

## ETHICS STATEMENT AND REPRODUCIBILITY GUIDELINES

In this work, we used well-established, publicly available datasets to train and evaluate our architecture. While these datasets and pre-trained models provide a valuable foundation for research, we acknowledge the potential for inherent biases that may not fully represent diverse real-world scenarios. We have taken every precaution to ensure that our experiments are conducted responsibly, with no intention of causing harm or perpetuating any biases present in the data. Furthermore, we declare no conflicts of interest in the execution or reporting of this research. Our objective is to present the findings in a transparent manner and contribute positively to the broader research community.

To ensure the reproducibility of our experiments, we have provided the code and detailed instructions on how to run the experiments. The general configuration of our model is described in Sect. 4, with additional technical details outlined in Sect. A.1 of the Appendix. Moreover, we have employed random seed initialization to ensure consistency across runs. The complete codebase, models, and pre-trained weights are available on GitHub [1] to facilitate further research and replication.

## ACKNOWLEDGMENTS

This work was supported by the "Knowledge Foundation" (KK-stiftelsen).

---

[1]https://github.com/bracca95/CAMeLU.git

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

# A  APPENDIX

## A.1  EXPERIMENTAL DETAILS

**Datasets.** For training CAMeLU, we use ImageNet-964, which is a variant of the original ImageNet-1k dataset (Deng et al., 2009) where classes belonging to the validation and test splits of *mini*ImageNet (Ravi & Larochelle, 2016) are removed. This results in a total of $1\,234\,487$ images for training the model compared to the $1\,281\,167$ in the original ImageNet-1k dataset. When a multi-dataset approach is utilized for training CAMeLU (see Appendix A.3), MSCOCO (Lin et al., 2014) and Fungi (Schroeder & Cui, 2018) are loaded into the program and used together with ImageNet-964 for creating the whole training dataset. MSCOCO is a dataset originally proposed for object detection, where each image is assigned to $G$ classes corresponding to the $G$ objects present in it. To use it for image classification, we replicate each image $G$ times and we assign to each of them one of the $G$ classes. In this way, we obtain a dataset with $117\,266$ images for training, and we rely on the fact that the transformer is capable of applying self-attention to the object of the class in question. Fungi is a fine-grained dataset with a size of only $64\,307$, which is two orders of magnitude smaller than ImageNet-964 and MSCOCO. For evaluation and for in-domain training of the baselines, we use *mini*ImageNet, CIFAR-fs, CUB, Aircraft, and Meta-iNat. *mini*ImageNet is split into train/validation/test using the splits proposed in Ravi & Larochelle (2016), resulting in $38\,400$ images for training, 9600 for validation, and $12\,000$ for testing. The same number of images are also present in CIFAR-fs, and the splits follow the work in Bertinetto et al. (2019). CUB and Aircraft, instead, are two fine-grained datasets with a smaller size compared to the others. CUB (Wah et al., 2011) consists of 8239 in the training set, 1779 in the validation set, and 1770 in the test set, while Aircraft has respectively 7000/1500/1500 images in the train/validation/test sets (Triantafillou et al., 2019). Finally, Meta-iNat (Wertheimer & Hariharan, 2019) consists of $243\,986$ images split into 1135 classes, with 227 reserved for testing. All images are resized to $224 \times 224$ and normalized with zero mean and unit variance before input into the model. For the UML baselines, due to the smaller model size utilized in the experiments, images are resized to $84 \times 84$, as suggested in the original papers (Khodadadeh et al., 2019; Hsu et al., 2018; Lee et al., 2020; Jang et al., 2022).

**CAMeLU.** The architecture used for CAMeLU consists of a fixed pre-trained feature extractor, a class encoder, and a transformer encoder. The feature extractor is pre-trained using ResNet-50 on ImageNet-964 following the same architecture and hyperparameters in He et al. (2016). The class encoder is a single learnable layer with a dimensionality of 256 and initialized with Kaiming initialization (He et al., 2015). Image embeddings are concatenated with class embeddings before being fed into the transformer encoder. This results in a vector with a total length of 2304, composed of 2048 features from the image embeddings and 256 from the label embeddings. When ablating the feature extractor with CLIP (Radford et al., 2021) in Appendix A.4, a ViT-B/16 encoder architecture is utilized and downloaded from the Hugging Face website (Wolf et al., 2019). The pre-training is performed using a large dataset with 400 million (image, text) pairs (Radford et al., 2021), and it is fixed during the training phase of CAMeLU. The output size of the image embedding is reduced to 1024, with 768 features from the image embedding, which results in a reduced memory complexity compared to ResNet-50. The transformer encoder comprises 8 encoder layers. Each layer consists of 8 attention heads and an MLP with a reversed bottleneck of 3072 (with GeLU activation function). A projection layer completes the model architecture to map the transformer output to a class prediction. This architecture enables us to store the entire model in an Nvidia GeForce RTX 3070 Ti Laptop GPU with 8GB of VRAM, while a further reduction of memory can be achieved by utilizing CLIP on ViT-B/16 as feature extractor, which requires only 4 GB of VRAM to store the entire model.

For the task creation mechanism, 3 augmentation functions are selected from a list comprising cropping, rotation, horizontal flip, grayscale, color jittering, gaussian blur, and random affine transformation. The exact parameters used in our experiments for each augmentation function are detailed in Tab. 4. For the query set, an additional pixel-level mixing strategy with $\lambda \sim Beta(\alpha, \beta)$ with $\alpha = 1, \beta = 1$ and $\lambda \in (0, 0.5)$ is utilized. More details about this selection choice can be found in Appendix A.6.

The training of CAMeLU is performed for 100 epochs, with 500 episodes each, using the Adam optimizer with an initial learning rate of $10^{-5}$ and a warmup cosine scheduler with 1500 warmup steps and a final learning rate of $10^{-6}$. For the evaluation, instead, a single forward pass is performed and the accuracy between the output and the true label is calculated on the query set of each given

Table 4: Complete list of transformations used for generating the support set of each task in CAMeLU. The names of the augmentations are taken from the *torchvision* library in Python.

| Augmentation | Parameters |
|---|---|
| *RandomResizedCrop* | image size $= 224$, scale $= (0.2, 0.8)$, ratio$= (0.75, 1.33)$ |
| *RandomRotation* | degrees $= 60$, probability $= 1.0$ |
| *RandomHorizontalFlip* | probability $= 1.0$ |
| *Grayscale* | output channels $= 3$ |
| *ColorJitter* | brightness $= 0.2$, contrast $= 0.2$, saturation $= 0.2$, hue $= 0.2$ |
| *GaussianBlur* | kernel size $= 3$, sigma $= (0.1, 2.0)$ |
| *RandomAffine* | degrees $= 0$, shear $= [-45, 45, -45, 45]$ |

task. Results are then averaged across 500 tasks, and the mean and standard deviation across three complete runs (consisting of training and evaluation) of the algorithm are used in our experiments.

**Baselines.** We compare our results with UML methods, an unsupervised few-shot learning method, a supervised meta-learning method, and two SSL methods. For the UML baselines, we consider CACTUs-MAML (Hsu et al., 2018), CACTUs-ProtoNet (Hsu et al., 2018), UMTRA (Khodadadeh et al., 2019), Meta-GMVAE (Lee et al., 2020), and PsCo (Jang et al., 2022). All these methods are evaluated in-domain, i.e., using the same dataset for training and evaluation, to adhere to the setting proposed in the original papers. Only PsCo is also extended to the cross-domain scenario that we discuss in this paper. All methods are trained for 100 epochs, using the parameters reported in the original papers (Khodadadeh et al., 2019; Hsu et al., 2018; Lee et al., 2020; Jang et al., 2022), and evaluated with 100 adaptation steps on each task when required by the model. When evaluated in-domain, all approaches use a Conv5 architecture consisting of 5 convolutional layers with 64 filters and a kernel size of 3, followed by batch normalization, ReLU non-linearity, max pooling, and a classifier head. The only exception is Meta-GMVAE. For this method, the authors trained a Conv5 feature extractor with SimCLR and input the learned features into a variational autoencoder (VAE) (Lee et al., 2020). Due to time limitations and the computational resources required to train a model with SimCLR, in our experiments, we used a feature extractor consisting of a pre-trained version of SimCLR on ResNet-50 (Borovec et al., 2022) using ImageNet-1k, followed by a projection layer fine-tuned for 100 steps to the training dataset, as done for the SSL baselines. This approach results in a better performance than the one reported in the original paper (Lee et al., 2020) (see Tab. 1), likely due to the improved ability of the feature extractor to extract meaningful features. For PsCo, when evaluated on a cross-domain setting, we utilized the ResNet-50 architecture trained on ImageNet-964 to avoid data leakage, and we then applied the model to the test domain using 100 adaptation steps to it. We also included BECLR (Poulakakis-Daktylidis & Jamali-Rad, 2024) in our comparison utilizing the same hyperparameters and model architectures proposed in the original paper, given the importance of hyperparameter choice for final performance. Specifically, the ResNet-50 feature extractor was trained only on *mini*ImageNet and evaluated on cross-domain scenarios in Tab. 2.

To compare our results with CAML (Fifty et al., 2024), the same architecture and hyperparameter of our approach are applied to this method. This results in a lower performance for CAML compared to the original paper (Fifty et al., 2024), as the reduced size of the model and the different feature extractor (ResNet-50 instead of ViT-CLIP), but it guarantees fair comparisons and lets us train the model with the available computational resources.

We also provide a comparison with two SSL approaches - SimCLR (Chen et al., 2020) and SwAV (Caron et al., 2020). The details for training and evaluation are provided in Sect. 4.6.

## A.2 IN-CONTEXT LEARNING ANALYSIS

To verify the contribution of the in-context learner in CAMeLU, we examine the embedding space learned during inference, both after the feature extractor and the transformer encoder. Fig. 5 presents a t-SNE visualization of a single test task, where clusters represent the embeddings of the support classes. For simplicity, we illustrate a 5-way 5-shot task with one query sample for each dataset, and we report the Euclidean distance between the query and the centroid of each class. As the dataset

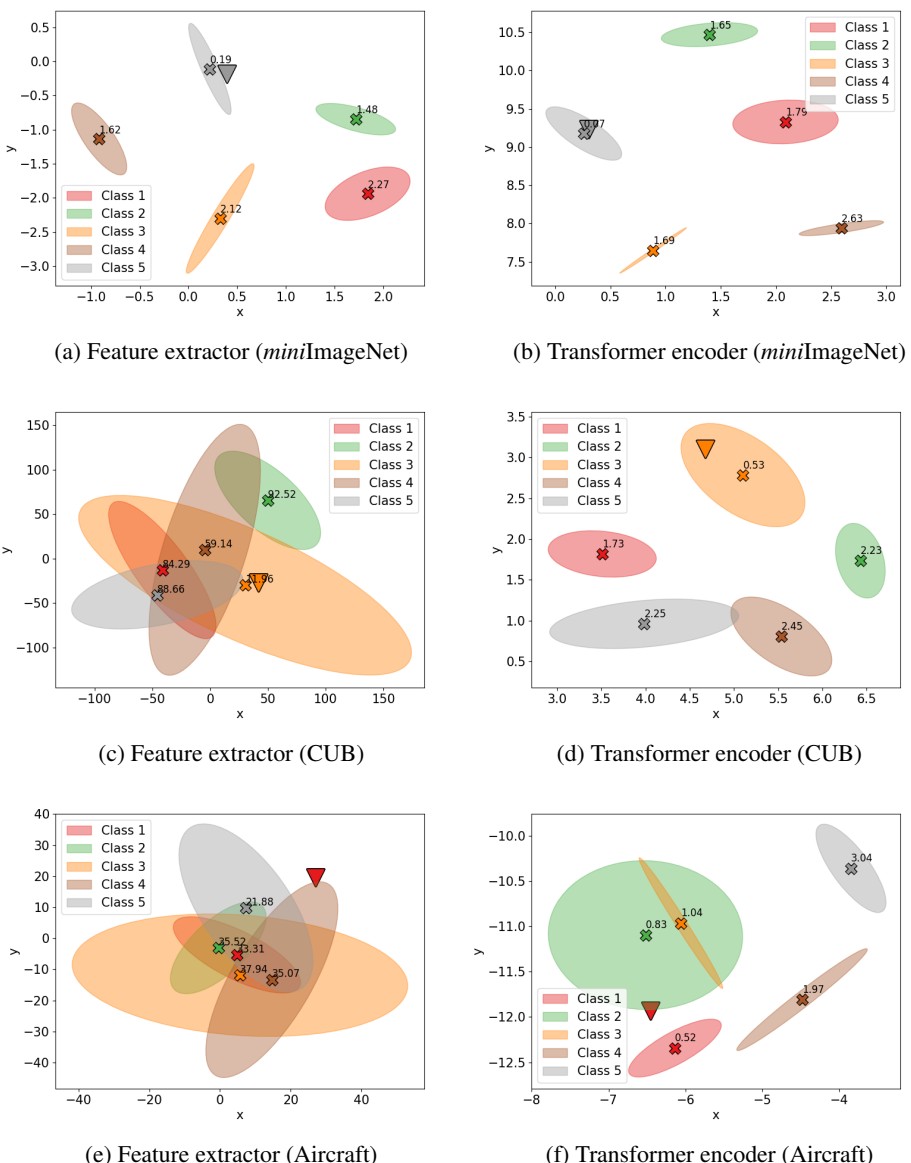

Figure 5: Visualization of clustered embeddings obtained with CAMeLU after the fixed feature extractor (left) and the transformer encoder (right) across different datasets. The plots represent 5-way 5-shot tasks during inference. Crosses indicate the centroids for each class, triangles represent the query sample embeddings, and the numbers denote the Euclidean distances between the query and each class centroid. The plots are obtained using t-SNE (Van der Maaten & Hinton, 2008) with a perplexity equals to 9.

Table 5: Comparison of CAMeLU and CAML when trained on the single ImageNet-964 dataset (IN-964) and on a multi-dataset (`mds`) consisting of ImageNet-964 + MSCOCO + Fungi. Results show the mean and standard deviations for the 5-way 1-shot and the 5-way 5-shot settings across three complete runs of the algorithms.

| | *mini*ImageNet | | CIFAR-fs | | CUB | | Aircraft | | Meta-iNat | |
|---|---|---|---|---|---|---|---|---|---|---|
| | 5w1s | 5w5s | 5w1s | 5w5s | 5w1s | 5w5s | 5w1s | 5w5s | 5w1s | 5w5s |
| CAMeLU (IN-964) | 76.51 ± 0.79 | 92.14 ± 0.30 | 61.79 ± 0.59 | 80.43 ± 0.21 | 65.52 ± 0.37 | 80.35 ± 0.63 | 33.17 ± 0.94 | 39.11 ± 1.97 | 57.27 ± 0.39 | 75.45 ± 0.42 |
| CAMeLU (mds) | 76.56 ± 0.36 | 91.82 ± 0.21 | 62.41 ± 0.70 | 80.36 ± 0.32 | 65.35 ± 0.70 | 79.78 ± 0.38 | 32.54 ± 0.41 | 37.87 ± 0.59 | 57.36 ± 0.33 | 75.40 ± 0.29 |
| CAML (IN-964) | 81.75 ± 0.18 | 92.31 ± 0.11 | 59.44 ± 0.63 | 75.27 ± 0.77 | 54.63 ± 1.78 | 66.81 ± 3.12 | 28.92 ± 0.37 | 32.06 ± 0.43 | 50.86 ± 0.50 | 67.07 ± 0.39 |
| CAML (mds) | 81.90 ± 0.54 | 92.93 ± 0.33 | 63.08 ± 0.43 | 79.73 ± 0.63 | 56.85 ± 1.92 | 69.43 ± 1.85 | 28.36 ± 2.26 | 31.56 ± 2.14 | 54.72 ± 0.63 | 70.50 ± 0.67 |

complexity increases, we observe greater variation between the embeddings learned from the fixed feature extractor and those refined by the transformer encoder. For instance, with the *mini*ImageNet dataset, the feature extractor alone is able to recognize the class of the query, clustering the query sample with the support samples belonging to the same class. However, in more challenging datasets such as CUB and Aircraft, the embedding space after the feature extractor appears more sparse, reflected by the large Euclidean distance between the query sample and the centroid of each class. In contrast, the transformer encoder significantly improves the representation, producing more compact and well-separated clusters, underscoring its crucial role in CAMeLU. For instance, in the Aircraft dataset, the query would be misclassified as class 5 (in grey) based on the feature extractor alone, but it is correctly classified after passing through the transformer. This highlights the role of the transformer encoder in updating the support and query representations based on the context of the task, not only the image context, improving classification accuracy.

### A.3 MULTI-DATASET TRAINING

We conduct additional experiments to evaluate the performance of CAMeLU when trained on a large-scale dataset. As our focus is on cross-domain classification through in-context learning, we hypothesize that training on a dataset spanning various concepts could enhance classification performance, as suggested in Min et al. (2022) and Fifty et al. (2024). To test this hypothesis, we combine three training datasets with varying levels of granularity: ImageNet-964 (Deng et al., 2009), MSCOCO (Lin et al., 2014), and Fungi (Schroeder & Cui, 2018). During each training episode, a dataset is uniformly sampled, and $N$ data points are extracted. These samples are then utilized in our method as described in Sect. 3. Table 5 presents the results for CAMeLU and the supervised CAML method. Notably, training with a combination of multiple datasets (denoted as `mds` in Tab. 5) yields improved performance for CAML (supervised) compared to training solely on ImageNet-964, likely due to the increased variability in the sampled tasks. However, CAMeLU does not exhibit a similar performance boost, as its task creation mechanism already introduces substantial variability, reducing the benefit of additional dataset diversity. Moreover, the large size of ImageNet-964 (around $80\%$ of the overall dataset), leads to its more frequent selection compared to the other datasets and limits the potential for performance gains from the other datasets. Consequently, to optimize computational resources and time, we conducted all the other experiments by training solely on ImageNet-964.

### A.4 ABLATION STUDIES - FEATURE EXTRACTOR

To assess the impact of the pre-trained feature extractor, we evaluate CAMeLU with various extractors pre-trained using different strategies. In particular, we compare the effectiveness of a ResNet-50 encoder pre-trained in a supervised manner both on ImageNet-1k and on ImageNet-964, a ResNet-50 pre-trained on ImageNet-1k using two SSL strategies, i.e., SimCLR (Chen et al., 2020) and SwAV (Caron et al., 2020), and a ViT-B/16 architecture pre-trained with CLIP (Radford et al., 2021) on a large dataset with 400 million (image, text) pairs. The extractors were downloaded from the Hugging Face (Wolf et al., 2019) and PyTorch Lightning Bolts (Borovec et al., 2022) websites, except for the ResNet-50 architecture pre-trained on ImageNet-964. Results in Tab. 6a demonstrate that the models pre-trained on ImageNet-1k exhibit significantly higher performance on *mini*ImageNet, compared to other datasets, primarily due to the data leakage issue described in Sect. 4.2. This issue is mitigated by pre-training the ResNet-50 architecture on ImageNet-964, which results in a drop in the performance on *mini*ImageNet and CIFAR-fs due to the removal of data leakage, making it

Table 6: Ablation study of feature extractors utilized in CAMeLU. The feature extractors include ResNet-50 pre-trained on ImageNet-964 (ResNet50 (IN-964)), ResNet-50 pre-trained on ImageNet-1k (ResNet50 (IN-1k)), ResNet-50 pre-trained on ImageNet-1k with SimCLR and SwAV, as well as a ViT-B/16 architecture pre-trained with CLIP. All models are trained using CAMeLU on (a) ImageNet-964 or (b) using a combination of ImageNet-964 + MSCOCO + Fungi (multi-dataset). The symbol † indicates results that are affected by data leakage. Results show the mean and standard deviations across three complete runs of the algorithms.

| Extractor | *mini*ImageNet | | CIFAR-fs | | CUB | | Aircraft | | Meta-iNat | |
|---|---|---|---|---|---|---|---|---|---|---|
| | 5w1s | 5w5s | 5w1s | 5w5s | 5w1s | 5w5s | 5w1s | 5w5s | 5w1s | 5w5s |
| ResNet-50 (IN-964) | $76.51 \pm 0.79$ | $92.14 \pm 0.30$ | $61.79 \pm 0.59$ | $80.43 \pm 0.21$ | $\mathbf{65.52 \pm 0.37}$ | $\mathbf{80.35 \pm 0.63}$ | $33.17 \pm 0.94$ | $39.11 \pm 1.97$ | $57.27 \pm 0.39$ | $75.45 \pm 0.42$ |
| ResNet-50 (IN-1k) | $\mathbf{78.17 \pm 1.69}^{\dagger}$ | $\mathbf{95.75 \pm 0.48}^{\dagger}$ | $66.02 \pm 0.78$ | $84.40 \pm 0.64$ | $60.69 \pm 1.13$ | $79.08 \pm 0.75$ | $33.23 \pm 0.70$ | $40.05 \pm 0.85$ | $56.21 \pm 0.43$ | $74.35 \pm 0.21$ |
| ResNet-50 (IN-1k) - SimCLR | $56.10 \pm 1.16^{\dagger}$ | $79.45 \pm 2.37^{\dagger}$ | $46.14 \pm 1.24$ | $63.03 \pm 2.73$ | $36.85 \pm 2.69$ | $50.34 \pm 3.22$ | $24.30 \pm 1.06$ | $27.25 \pm 2.21$ | $42.61 \pm 0.41$ | $58.95 \pm 0.72$ |
| ResNet-50 (IN-1k) - SwAV | $60.16 \pm 0.70^{\dagger}$ | $84.32 \pm 0.34^{\dagger}$ | $56.81 \pm 0.84$ | $75.49 \pm 1.76$ | $44.39 \pm 0.75$ | $60.44 \pm 0.18$ | $27.82 \pm 0.62$ | $34.56 \pm 1.67$ | $47.31 \pm 0.41$ | $65.84 \pm 0.09$ |
| ViT-B/16 - CLIP | $76.44 \pm 0.51$ | $91.96 \pm 0.31$ | $\mathbf{69.74 \pm 0.95}$ | $\mathbf{86.25 \pm 0.92}$ | $61.05 \pm 1.91$ | $75.17 \pm 2.77$ | $\mathbf{37.82 \pm 2.14}$ | $\mathbf{43.10 \pm 1.95}$ | $\mathbf{61.22 \pm 0.67}$ | $\mathbf{77.09 \pm 0.15}$ |

(a) ImageNet-964 training

| Extractor | *mini*ImageNet | | CIFAR-fs | | CUB | | Aircraft | | Meta-iNat | |
|---|---|---|---|---|---|---|---|---|---|---|
| | 5w1s | 5w5s | 5w1s | 5w5s | 5w1s | 5w5s | 5w1s | 5w5s | 5w1s | 5w5s |
| ResNet-50 (IN-964) | $76.56 \pm 0.36$ | $91.80 \pm 0.20$ | $62.28 \pm 0.69$ | $80.15 \pm 0.37$ | $65.06 \pm 0.82$ | $79.27 \pm 1.22$ | $31.89 \pm 1.43$ | $37.13 \pm 1.67$ | $57.36 \pm 0.33$ | $75.04 \pm 0.49$ |
| ResNet-50 (IN-1k) | $\mathbf{79.07 \pm 0.88}^{\dagger}$ | $\mathbf{96.44 \pm 0.16}^{\dagger}$ | $66.15 \pm 0.31$ | $84.90 \pm 0.42$ | $60.62 \pm 0.45$ | $79.26 \pm 0.20$ | $33.41 \pm 0.98$ | $41.23 \pm 1.14$ | $59.14 \pm 0.14$ | $74.31 \pm 0.51$ |
| ResNet-50 - SimCLR | $53.83 \pm 1.87$ | $78.10 \pm 1.94$ | $45.06 \pm 1.06$ | $61.90 \pm 0.33$ | $37.64 \pm 1.74$ | $51.40 \pm 1.70$ | $25.31 \pm 0.49$ | $28.87 \pm 0.99$ | $41.89 \pm 0.15$ | $58.87 \pm 0.36$ |
| ResNet-50 - SwAV | $58.82 \pm 0.34$ | $83.45 \pm 0.24$ | $57.33 \pm 0.57$ | $76.62 \pm 1.01$ | $44.79 \pm 0.24$ | $60.71 \pm 0.85$ | $27.30 \pm 0.77$ | $34.50 \pm 0.85$ | $47.18 \pm 0.35$ | $65.65 \pm 0.19$ |
| ViT-B/16 - CLIP | $77.92 \pm 1.89$ | $93.83 \pm 0.70$ | $\mathbf{78.04 \pm 0.91}$ | $\mathbf{91.88 \pm 0.45}$ | $\mathbf{74.08 \pm 1.81}$ | $\mathbf{88.86 \pm 2.33}$ | $\mathbf{49.21 \pm 2.46}$ | $\mathbf{58.97 \pm 2.74}$ | $\mathbf{67.95 \pm 1.25}$ | $\mathbf{82.59 \pm 1.05}$ |

(b) Multi-dataset training

comparable with the results on CLIP-ViT-B/16. For CLIP-ViT-B/16, however, thorough verification of potential data leakage was not possible due to the undisclosed nature of its training dataset. As such, these results should be interpreted with caution. CLIP-ViT-B/16 stands out as the best-performing method due to its dataset-agnostic nature and its ability to learn representations that generalize across a broad range of tasks. Furthermore, when training CAMeLU with a multi-dataset approach (Tab. 6b), as described in Appendix A.3, results for CLIP-ViT-B/16 improve further, highlighting its applicability also to datasets significantly different from those used for training (Radford et al., 2021). These findings demonstrate that CAMeLU's performance scales with the strength of the feature extractor, indicating potential for further investigation as more robust feature extractors become available. However, in this work, we decided to use the ResNet-50 architecture to guarantee a fair comparison with previous baselines.

## A.5 EVALUATION OF THE TASK CREATION MECHANISM

To assess the effectiveness of the task creation strategy employed in our proposed approach, we conduct a comparative analysis of CAMeLU's performance under two different task creation mechanisms. Specifically, we evaluate CAMeLU when tasks are generated solely using data augmentations for both the support and query sets, following a strategy similar to UMTRA (Khodadadeh et al., 2019), versus employing our proposed approach outlined in Sect. 3.1. By applying our task creation strategy, we generate more complex tasks, making the generalization problem harder and the in-context learner more robust (Chan et al., 2022; Singh et al., 2024). The results presented in Tab. 7a confirm our claim. Across all the datasets, our proposed strategy enhances the generalization on cross-domain datasets such as CUB, Aircraft, and Meta-iNat. This conclusion is further supported by Fig. 6, which shows the validation accuracy on *mini*ImageNet and CUB using the two mechanisms discussed before, along with the CLIP-ViT feature extractor described in Appendix A.4. These results confirm

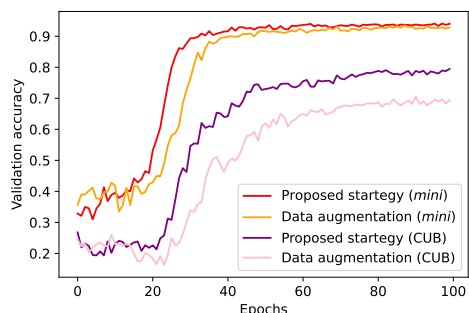

Figure 6: Validation accuracy on *mini*ImageNet (*mini*) and CUB while training CAMeLU with two different task creation mechanisms. Red and purple curves are obtained with our proposed strategy in Sect. 3.1, while orange and pink curves are obtained by applying only data augmentations based on image manipulations to generate the support and query samples. The training is performed using ImageNet-964.

Table 7: Ablation experiments of the proposed task creation mechanism by (a) generating tasks only using data augmentations for the support and query set on ImageNet-964 and (b) applying k-means clustering on the ResNet-50 embeddings on *mini*ImageNet. Results show the mean and standard deviations across three complete runs of the algorithms in the 5-way 1-shot and 5-way 5-shot scenarios.

| Method | *mini*ImageNet | | CIFAR-fs | | CUB | | Aircraft | | Meta-iNat | |
|---|---|---|---|---|---|---|---|---|---|---|
| | 5w1s | 5w5s | 5w1s | 5w5s | 5w1s | 5w5s | 5w1s | 5w5s | 5w1s | 5w5s |
| **ImageNet-964** | | | | | | | | | | |
| Augment | **78.20 ± 0.38** | 91.35 ± 0.35 | **64.30 ± 0.31** | **81.08 ± 0.23** | 62.19 ± 1.29 | 75.53 ± 1.52 | 31.90 ± 1.69 | 37.46 ± 1.71 | 56.46 ± 0.53 | 74.00 ± 0.80 |
| Proposed | 76.51 ± 0.79 | **92.14 ± 0.30** | 61.79 ± 0.59 | 80.43 ± 0.21 | **65.52 ± 0.37** | **80.35 ± 0.63** | **33.17 ± 0.94** | **39.11 ± 1.97** | **57.27 ± 0.39** | **75.45 ± 0.42** |

(a) Data augmentation vs. the proposed strategy

| Method | *mini*ImageNet | | CIFAR-fs | | CUB | | Aircraft | | Meta-iNat | |
|---|---|---|---|---|---|---|---|---|---|---|
| | 5w1s | 5w5s | 5w1s | 5w5s | 5w1s | 5w5s | 5w1s | 5w5s | 5w1s | 5w5s |
| *mini*ImageNet | | | | | | | | | | |
| k-means | 75.86 ± 1.27 | 89.57 ± 0.65 | 46.14 ± 1.96 | 62.19 ± 2.19 | 33.76 ± 3.58 | 40.39 ± 4.06 | 24.48 ± 3.34 | 27.11 ± 4.23 | 36.32 ± 1.68 | 47.66 ± 1.59 |
| Proposed | **75.99 ± 0.20** | **90.38 ± 0.21** | **61.25 ± 0.55** | **78.79 ± 0.21** | **60.60 ± 0.80** | **74.77 ± 1.70** | **31.39 ± 1.17** | **36.52 ± 0.88** | **55.60 ± 0.20** | **72.12 ± 0.35** |

(b) k-means clustering vs. the proposed strategy

Table 8: Evaluation of the proposed task creation strategy when applied to build pseudo-tasks on top of MAML. The results are compared with other MAML-based baselines for UML, such as UMTRA and CACTUs-MAML.

| Method | *mini*ImageNet | | CIFAR-fs | | CUB | | Aircraft | | Meta-iNat | |
|---|---|---|---|---|---|---|---|---|---|---|
| | 5w1s | 5w5s | 5w1s | 5w5s | 5w1s | 5w5s | 5w1s | 5w5s | 5w1s | 5w5s |
| CACTUs-MAML | **43.30** | **54.21** | **42.00** | **56.64** | 31.19 | 36.81 | 24.06 | 27.26 | 20.13 | 21.84 |
| UMTRA | 39.93 | 50.73 | 32.93 | 46.13 | 27.06 | 36.60 | 22.40 | 31.73 | 28.96 | 37.12 |
| Proposed + MAML | 34.04 | 46.13 | 37.02 | 52.00 | **31.22** | **41.54** | **26.12** | **34.34** | **30.94** | **42.43** |

that our task creation strategy is more robust, particularly in cross-domain evaluations, even when stronger feature extractors are utilized.

Additionally, we experimented with k-means clustering as an alternative to the proposed task creation strategy. Inspired by CACTUs (Hsu et al., 2018), we applied clustering on the embeddings generated by the feature extractor to generate pseudo-labels. The results are presented in Tab. 7b when training on *mini*ImageNet due to the high computational cost of k-means and indicate that our proposed mechanism generalizes better across domains. Furthermore, k-means clustering requires an insight into the number of classes in the training dataset, as choosing a high number of clusters would lead to a lack of samples per class, whereas a low number may hinder generalization. In contrast, CAMeLU does not rely on such assumptions, enhancing its robustness compared to clustering-based approaches.

Finally, we show that the benefits of our task creation strategy extend beyond CAMeLU. In Tab. 8 we apply our proposed mechanism to generate pseudo-tasks on top of MAML (Finn et al., 2017). This allows for a direct comparison with MAML-based approaches, such as UMTRA (Khodadadeh et al., 2019) and CACTUs-MAML (Hsu et al., 2018), by replacing their original task creation mechanisms. The increased performance of our strategy applied to the baselines demonstrates its superiority over previous task creation methods. It may be objected that CACTUs-MAML achieves higher performance on *mini*ImageNet and CIFAR-fs. However, this is caused by the use of a feature extractor pre-trained on ImageNet-1k, which introduces an unfair advantage by leaking information about the test data into the training phase. This performance gap narrows when the test distribution deviates from the training data (e.g., CIFAR-fs) and disappears for datasets with low correlation to ImageNet-1k, supporting our hypothesis. Indeed, on datasets that share low similarity with ImageNet-1k, our method consistently outperforms both UMTRA and CACTUs-MAML. While these results highlight the strength of our task creation strategy, the performance still remains significantly lower than CAMeLU, especially in cross-domain scenarios. This emphasizes the critical role of combining our robust task creation mechanism with the in-context learning capabilities of transformer-based architectures to achieve superior performance.

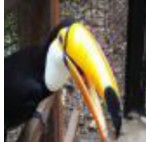 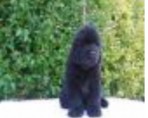 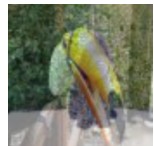 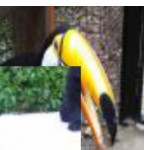

(a) Image $\tilde{x}_{n,j}$   (b) Random image $z_j$   (c) Query image $x_j^{(qr)}$ with pixel level mix   (d) Query image $x_j^{(qr)}$ with patch level mix

Figure 7: Visualization of a query image $\tilde{x}_{n,j}$ generated by mixing (a) the support image and (b) a randomly sampled image at (c) the pixel level or at (d) the patch level using $\lambda = 0.49$.

Table 9: $mSSIM$ values computed as the average between $SSIM(\tilde{x}_{n,j}, x_j^{(qr)})$ and $SSIM(z_j, x_j^{(qr)})$ when $x_j^{(qr)}$ is obtained using a pixel level or a patch level mixing strategy with $\lambda = 0.25$ and $\lambda = 0.49$.

|             | $\lambda = 0.25$ | $\lambda = 0.49$ |
|-------------|------------------|------------------|
| Pixel level | 0.60             | 0.61             |
| Patch level | 0.56             | 0.57             |

A.6   QUERY SAMPLES GENERATION STRATEGY

We also conducted additional experiments to validate the choice of utilizing Eq. 1 for generating query samples. In particular, we compare the results obtained as a linear combination of the augmented image $\tilde{x}_{n,j}$ with the randomly sampled image $z_j$ (pixel level), as in Eq. 1, and at the patch level, as in Yun et al. (2019). Specifically, for the latter, we randomly select a patch from $z_j$ with an area ratio proportional to $\lambda$, and we paste it into $\tilde{x}_{n,j}$. Fig. 7 illustrates an example of these two techniques by mixing two images sampled from ImageNet-964. As shown in Fig. 7c, merging the images at the pixel level results in a mixed image where some information from $\tilde{x}_{n,j}$ and $z_j$ is retained in every part of the image. Contrarily, in Fig. 7d, there is no information about $\tilde{x}_{n,j}$ in the lower left corner, forcing the network to attend only to the upper right part of the image to classify it with the same class as $\tilde{x}_{n,j}$. Therefore, we hypothesize that the pixel level strategy is more suitable for our approach as the goal is to attend to the whole image to extract robust features that allow the model to classify the query image with the same class as the support one while ensuring diversity between the two. To validate this, we utilized the Structural Similarity Index ($SSIM$) (Wang et al., 2004). $SSIM$ is used as a metric to measure the similarity between two given images based on three image features: luminance, contrast, and structure. Formally, considering $x$ and $y$ two given images, $SSIM$ is calculated as follows:

$$SSIM(x,y) = \left[\frac{2\mu_x\mu_y + c_1}{\mu_x^2 + \mu_y^2 + c_1}\right]^{\alpha} + \left[\frac{2\sigma_x\sigma_y + c_2}{\sigma_x^2 + \sigma_y^2 + c_2}\right]^{\beta} + \left[\frac{\sigma_{xy} + c_3}{\sigma_x\sigma_y + c_3}\right]^{\gamma} \tag{5}$$

where $\mu$ represents the mean of an image, $\sigma$ denotes the standard deviation, $c_1, c_2, c_3$ are constant values, and $\alpha, \beta, \gamma$ denote the relative importance of each metrics. By assuming $\alpha = \beta = \gamma = 1$ and $c_2 = c_2/2$, we get

$$SSIM(x,y) = \frac{(2\mu_x\mu_y + c_1)(2\sigma_{xy} + c_2)}{(\mu_x^2 + \mu_y^2 + c_1)(\sigma_x^2 + \sigma_y^2 + c_2)}. \tag{6}$$

Instead of applying the above formula all over the image at once, (Wang et al., 2004) proposed a local variant that consists of computing the $SSIM$ index locally and averaging these values to obtain the global $SSIM$ value. For our purpose, we computed this metric between each of the two images used for the generation, i.e., $\tilde{x}_{n,j}$ and $z_j$, and the resulting mixed image, i.e., $x_j^{(qr)}$. We then average the results, obtaining an indicator, denoted as $mSSIM$, of how similar the query image is with the images used for the generation, or in other words, how much local information is retained from $\tilde{x}_{n,j}$ and $z_j$ into $x_j^{(qr)}$. Results are shown in Tab. 9 for $\lambda = 0.25$ and $\lambda = 0.49$, confirming the hypothesis that, even for high $\lambda$ values, mixing at the pixel level guarantees more information retained across the whole image compared to using a patch level strategy. This is also confirmed by the results in Tab. 10, which shows a decrease in the performance when CAMeLU is trained with the patch level strategy for query generation.

We also ablate the values of the $\alpha$ and $\beta$ parameters used in the $Beta$ distribution from which $\lambda$ is sampled. Tab. 10 presents the results for different values of $\alpha$ and $\beta$ when $\lambda \sim Beta(\alpha, \beta)$ and $\lambda \in (0, 0.5)$. The results indicate that the optimal choice for CAMeLU is to select $\alpha = 1, \beta = 1$,

Table 10: Accuracy results of CAMeLU when trained with different strategies for generating the query samples. *Pixel level mix* refers to the scenario where query samples are generated with Eq. 1, while *patch level mix* refers to a strategy similar to the one proposed in Yun et al. (2019). Results are reported in the 5-way 5-shot scenario with $\lambda \sim Beta(\alpha, \beta)$ and different $\alpha$ and $\beta$ values. Results show the mean and standard deviations across three complete runs of the algorithms.

| | *mini*ImageNet | CIFAR-fs | CUB | Aircraft |
|---|---|---|---|---|
| Pixel level mix | | | | |
| $\alpha = 0.1, \beta = 0.1$ | $90.68 \pm 0.67$ | $75.88 \pm 2.08$ | $76.31 \pm 2.44$ | $35.31 \pm 2.57$ |
| $\alpha = 0.5, \beta = 0.5$ | $91.34 \pm 0.24$ | $77.70 \pm 0.73$ | $79.52 \pm 0.73$ | $37.56 \pm 1.92$ |
| $\alpha = 1, \beta = 1$ | $\mathbf{92.14 \pm 0.30}$ | $\mathbf{80.43 \pm 0.21}$ | $\mathbf{80.35 \pm 0.63}$ | $\mathbf{39.11 \pm 1.97}$ |
| $\alpha = 2, \beta = 5$ | $90.68 \pm 1.02$ | $77.00 \pm 1.40$ | $79.62 \pm 2.50$ | $38.02 \pm 1.85$ |
| $\alpha = 5, \beta = 5$ | $90.63 \pm 0.30$ | $78.12 \pm 0.15$ | $79.71 \pm 0.42$ | $37.57 \pm 0.11$ |
| Patch level mix | | | | |
| $\alpha = 1, \beta = 1$ | $91.25 \pm 0.50$ | $77.80 \pm 0.45$ | $76.12 \pm 1.06$ | $33.60 \pm 1.27$ |

Table 11: Computational and time complexity of CAMeLU in comparison with PsCo. The comparison is performed considering the time required for the task creation, the training time (expressed in time per epoch), the inference time on a single task, and the GPU and CPU memory usage during training and inference.

| | Time task construction (ms) | Training time (ms/epoch) | Inference time (ms/task) |
|---|---|---|---|
| PsCO | 20772 | 4613656 | 605 |
| CAMeLU | 1376 | 153000 | 57 |

| | GPU training (MiB) | CPU training (MiB) | GPU inference (MiB) | CPU inference (MiB) |
|---|---|---|---|---|
| PsCO | 43904 | 20904 | 1630 | 2061 |
| CAMeLU | 6250 | 2588 | 3224 | 1667 |

which appears to be a uniform distribution. Additionally, $\alpha = 2, \beta = 5$ also yields comparable results, highlighting the importance of selecting a sufficiently small $\lambda$ to ensure the incorporation of sufficient information from $\tilde{x}_{n,j}$ into $x_j^{(qr)}$ facilitating the model's ability to classify the latter with the same class as $x_{n,i}^{(sp)}$.

### A.7 COMPUTATIONAL COMPLEXITY AND RESOURCES USAGE

In this Sect., we analyze the computational and time complexity of CAMeLU and we compare it with PsCo Jang et al. (2022). Tab. 11 presents the time required for task generation, model training, and inference, along with GPU and CPU memory usage. The results demonstrate that CAMeLU is not only faster than PsCo but also significantly more memory efficient. Notably, CAMeLU requires only 57ms for task inference, making it particularly suitable for real-time applications.

The computational complexity of CAMeLU is primarily attributed to the transformer architecture, which is known for its computational demands due to the self-attention mechanism. The transformer model has a computational complexity of $O(n^2 \cdot d)$ per layer (Vaswani et al., 2017), where $n$ is the sequence length and $d$ is the hidden dimension. In our context, $n$ includes both the support samples and the query sample. Consequently, the total computational complexity for evaluating $Q$ queries is $O(Q \cdot (NK + 1)^2 \cdot d)$, where $N$ is the number of classes, $K$ the number of shots, $NK + 1$ indicates one query per input sequence, and $Q$ is the total number of queries. This results in a quadratic complexity in the number of support samples which can be computationally demanding. However, we have demonstrated in Tab. 1 that CAMeLU achieves good performance even with only $K = 1$ support sample per class. Additionally, further experiments with only one query sample per episode

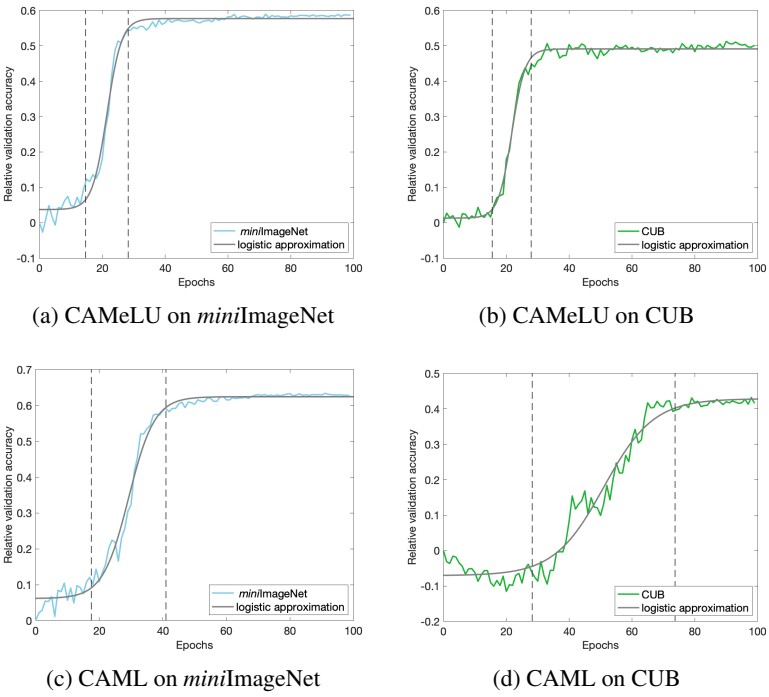

Figure 8: Comparison of logistic function approximations and phase boundaries for learning and generalization phases in CAMeLU and CAML for *mini*ImageNet and CUB datasets.

and one support sample per class (i.e., $N = 5$, $K = 1$, $Q = 1$) yield results of $80.74 \pm 0.65$ for *mini*ImageNet, $63.07 \pm 1.14$ for CIFAR-fs, $54.84 \pm 1.41$ for CUB, $30.32 \pm 0.76$ for Aircraft, and $55.76 \pm 0.08$ for Meta-iNat. These results are comparable to those reported in Tab. 1 for the 5-ways 1-shot scenario using 25 queries, highlighting that a single query is sufficient for good performance, as also demonstrated in CAML.

## A.8 QUANTITATIVE ANALYSIS OF LEARNING PHASES

To quantitatively assess the number of epochs required to enter the generalization phase, we propose to approximate the validation accuracy curves with the generalized logistic function

$$f(x) = a + \frac{d - a}{1 + e^{-b(x - x_0)}} = a + \frac{d - a}{1 + ce^{-bx}}, \tag{7}$$

where parameters $a, b, c, d$ are responsible for particular features of the logistic function. Parameters $a$ and $d$ indicate the lower and, respectively, the upper asymptote. Parameter $b$ is the logistic growth rate, and finally, parameter $c = e^{bx_0}$ is related to the inflection point $x_0$ at which the maximum growth of the function occurs. To find the best fitting logistic curve we use a standard regression function. The logistic function is strictly increasing, thus the derivative, which is given by $f'(x) = \frac{bc(d-a)e^{-bx}}{(1+ce^{-bx})^2}$, is always positive. The derivative firstly increases (from values close to zero), and after reaching its maximum value at the inflection point, it decreases. To determine the bounds for reaching the learning and generalization phases, we find the values for which the derivative is equal to a given fraction of the maximum possible growth rate. After testing several cases, the results show that the choice of this threshold does not affect the relation between the phases' boundaries. Therefore, we decided to conduct our analysis for $20\%$ of maximum growth rate.

Figures 8a and 8b show the results for CAMeLU on the *mini*ImageNet and CUB datasets, respectively. For *mini*ImageNet, we obtain the approximation function to be $f(x) = 0.04 + \frac{0.54}{1+9636e^{-0.43x}}$. Moreover, the number of epochs where the learning phase begins is 15 and the number of epochs where the generalization phase begins is 29. On the other hand, for CUB dataset, the approximation function is $f(x) = 0.01 + \frac{0.48}{1+25530e^{-0.47x}}$, and the number of epochs where the following phases begin is 16 and 28.

Figures 8c and 8d show the results for CAML on the *mini*ImageNet and the CUB datasets, respectively. Similarly, we obtain the approximation function for the *mini*ImageNet to be $f(x) = 0.06 + \frac{0.56}{1+1326e^{-0.25x}}$. and the number of epochs where the learning phase and the generalization phase begin is $18$ and $42$. Finally, for the CUB dataset, the approximation function is $f(x) = -0.07 + \frac{0.50}{1+648e^{-0.13x}}$, and the number of epochs where the following phases begin is $29$ and $74$.

Remarkably, CAML requires more training time to reach the generalization phase than CAMeLU. This difference likely arises from CAMeLU's task creation mechanism, which generates tasks with high cross-task variance. This strategy acts as a form of task augmentation, facilitating quicker generalization to unseen tasks.

## A.9 LIMITATIONS

Despite the promising results demonstrated by our novel UML approach on several datasets, some limitations remain. Its applicability and robustness in real-world scenarios with diverse and noisy data remain to be thoroughly evaluated. In real-world applications, data can be incomplete, mislabeled, or drawn from significantly different distributions, leading to potential degradation in the model's performance in the presence of noisy or corrupted data. Additionally, the feature extractor used in CAMeLU is pre-trained in a supervised manner. While the pre-training dataset is independent of the data seen at inference, replacing it with an extractor pre-trained using an SSL strategy could make the pipeline fully unsupervised, albeit at the price of performance degradation. Lastly, the proposed approach is designed to handle a fixed number of classes (ways) per task during training and testing, requiring knowing the value of $N$ in advance. Modifications to the task creation and training process would be necessary to extend our approach to handle an arbitrary number of ways.

## A.10 COMPLETE RESULTS WITH STANDARD DEVIATIONS

Table 12: Performance comparison on *mini*ImageNet, CIFAR-fs, CUB, Aircraft, and Meta-iNat datasets for 5-way 1-shot and 5-way 5-shot scenarios. Cross-domain approaches are trained using ImageNet-964 and a ResNet-50 feature extractor. The symbol † indicates results that are affected by data leakage. The bold font highlights the best performing UML approach for each setting. Results show the mean and standard deviations across three complete runs of the algorithms. This table refers to Tab. 1 in Sect. 4.3.

| Method | *mini*ImageNet | | CIFAR-fs | | CUB | | Aircraft | | Meta-iNat | |
|---|---|---|---|---|---|---|---|---|---|---|
| | 5w1s | 5w5s | 5w1s | 5w5s | 5w1s | 5w5s | 5w1s | 5w5s | 5w1s | 5w5s |
| **In-Domain** | | | | | | | | | | |
| CACTUs-MAML | $43.30 \pm 0.29$ | $54.21 \pm 1.00$ | $42.00 \pm 1.47$ | $56.64 \pm 0.49$ | $31.19 \pm 0.37$ | $36.81 \pm 0.68$ | $24.06 \pm 0.78$ | $27.26 \pm 0.04$ | $20.13 \pm 0.44$ | $21.840 \pm 0.14$ |
| CACTUs-ProtoNet | $48.85 \pm 0.69$ | $62.52 \pm 0.71$ | $50.90 \pm 0.46$ | $64.52 \pm 0.94$ | $33.93 \pm 0.37$ | $44.41 \pm 1.31$ | $26.27 \pm 0.28$ | $30.88 \pm 0.51$ | $27.30 \pm 0.12$ | $29.08 \pm 0.13$ |
| UMTRA | $39.93 \pm 1.15$ | $50.73 \pm 0.67$ | $32.93 \pm 1.68$ | $46.13 \pm 2.81$ | $27.06 \pm 1.41$ | $36.6 \pm 2.43$ | $22.40 \pm 3.42$ | $31.73 \pm 2.25$ | $28.96 \pm 0.32$ | $37.12 \pm 0.21$ |
| Meta-GMVAE | $55.38 \pm 0.90^\dagger$ | $65.10 \pm 0.64^\dagger$ | $52.02 \pm 0.88$ | $64.18 \pm 0.62$ | $33.59 \pm 0.63$ | $39.09 \pm 0.57$ | $24.83 \pm 0.51$ | $27.60 \pm 0.52$ | $34.22 \pm 0.58$ | $40.23 \pm 0.54$ |
| PsCo | $47.29 \pm 0.41$ | $64.85 \pm 0.38$ | $42.21 \pm 0.46$ | $62.92 \pm 0.44$ | $33.09 \pm 0.44$ | $51.02 \pm 0.42$ | $26.19 \pm 0.30$ | $38.80 \pm 0.38$ | $36.97 \pm 0.39$ | $55.88 \pm 0.41$ |
| **Cross-Domain** | | | | | | | | | | |
| PsCo | $67.89 \pm 0.48$ | $90.17 \pm 0.23$ | $53.34 \pm 0.49$ | $76.22 \pm 0.40$ | $43.35 \pm 0.47$ | $70.19 \pm 0.46$ | $29.87 \pm 0.36$ | $38.20 \pm 0.39$ | $46.21 \pm 0.44$ | $70.05 \pm 0.45$ |
| **CAMeLU** | $\mathbf{76.51 \pm 0.79}$ | $\mathbf{92.14 \pm 0.30}$ | $\mathbf{61.79 \pm 0.59}$ | $\mathbf{80.43 \pm 0.21}$ | $\mathbf{65.52 \pm 0.37}$ | $\mathbf{80.35 \pm 0.63}$ | $\mathbf{33.17 \pm 0.94}$ | $\mathbf{39.11 \pm 1.97}$ | $\mathbf{57.27 \pm 0.39}$ | $\mathbf{75.45 \pm 0.42}$ |
| CAML (supervised) | $81.75 \pm 0.18$ | $92.31 \pm 0.11$ | $59.44 \pm 0.63$ | $75.27 \pm 0.77$ | $54.63 \pm 1.78$ | $66.81 \pm 3.12$ | $28.92 \pm 0.37$ | $32.06 \pm 0.43$ | $50.86 \pm 0.50$ | $67.07 \pm 0.39$ |

Table 13: Accuracy results obtained training PsCo, BECLR, and CAMeLU with a small-scale dataset, namely *mini*ImageNet, denoted as (*mini*) in the table. Results show both in-domain performance (on the test set of *mini*ImageNet) and cross-domain performance on CIFAR-fs, CUB, Aircraft, and Meta-iNat. The mean and standard deviation across three complete runs of the algorithms. This table refers to Tab. 2 in Sect. 4.5.

| | *mini*ImageNet | | CIFAR-fs | | CUB | | Aircraft | | Meta-iNat | |
|---|---|---|---|---|---|---|---|---|---|---|
| | 5w1s | 5w5s | 5w1s | 5w5s | 5w1s | 5w5s | 5w1s | 5w5s | 5w1s | 5w5s |
| PsCo (*mini*) | $47.29 \pm 0.41$ | $64.85 \pm 0.38$ | $42.21 \pm 0.46$ | $62.92 \pm 0.44$ | $33.09 \pm 0.44$ | $51.02 \pm 0.42$ | $26.19 \pm 0.30$ | $\mathbf{38.80 \pm 0.38}$ | $36.97 \pm 0.39$ | $55.88 \pm 0.41$ |
| BECLR (*mini*) | $\mathbf{81.04 \pm 1.24}$ | $87.88 \pm 0.66$ | $57.05 \pm 1.58$ | $72.82 \pm 0.95$ | $42.47 \pm 1.30$ | $58.03 \pm 1.12$ | $27.48 \pm 0.83$ | $38.46 \pm 0.95$ | $49.87 \pm 1.35$ | $65.05 \pm 1.07$ |
| CAMeLU (*mini*) | $75.99 \pm 0.20$ | $\mathbf{90.38 \pm 0.21}$ | $\mathbf{61.25 \pm 0.55}$ | $\mathbf{78.79 \pm 0.21}$ | $\mathbf{60.60 \pm 0.80}$ | $\mathbf{74.77 \pm 1.70}$ | $\mathbf{31.39 \pm 1.17}$ | $36.52 \pm 0.88$ | $\mathbf{55.60 \pm 0.20}$ | $\mathbf{72.12 \pm 0.35}$ |

Table 14: Comparison between CAMeLU and SSL approaches for the 5-way 1-shot and 5-way 5-shot scenario on *mini*ImageNet, CIFAR-fs, CUB, Aircraft, and Meta-iNat. The symbol † indicates results that are affected by data leakage. Results show the mean and standard deviations across three complete runs of the algorithms. This table refers to Tab. 3 in Sect. 4.6.

| Method | *mini*ImageNet | | CIFAR-fs | | CUB | | Aircraft | | Meta-iNat | |
|---|---|---|---|---|---|---|---|---|---|---|
| | 5w1s | 5w5s | 5w1s | 5w5s | 5w1s | 5w5s | 5w1s | 5w5s | 5w1s | 5w5s |
| SimCLR | $\mathbf{83.32 \pm 0.23}^{\dagger}$ | $94.86 \pm 0.61^{\dagger}$ | $64.52 \pm 0.69$ | $84.36 \pm 0.40$ | $47.35 \pm 0.53$ | $66.87 \pm 0.82$ | $29.36 \pm 0.90$ | $39.99 \pm 0.86$ | $52.44 \pm 0.47$ | $73.19 \pm 0.43$ |
| SwAV | $74.83 \pm 0.71^{\dagger}$ | $\mathbf{94.96 \pm 0.91}^{\dagger}$ | $\mathbf{66.97 \pm 0.15}$ | $\mathbf{87.14 \pm 0.10}$ | $47.84 \pm 0.31$ | $69.31 \pm 0.01$ | $30.33 \pm 0.31$ | $\mathbf{47.43 \pm 0.11}$ | $53.57 \pm 0.82$ | $74.53 \pm 0.92$ |
| **CAMeLU** | $76.51 \pm 0.79$ | $92.14 \pm 0.30$ | $61.79 \pm 0.59$ | $80.43 \pm 0.21$ | $\mathbf{65.52 \pm 0.37}$ | $\mathbf{80.35 \pm 0.63}$ | $\mathbf{33.17 \pm 0.94}$ | $39.11 \pm 1.97$ | $\mathbf{57.27 \pm 0.39}$ | $\mathbf{75.45 \pm 0.42}$ |

