# OpenReview forum: "Unsupervised Meta-Learning via In-Context Learning"
_ICLR.cc/2025/Conference — ICLR 2025 Poster_

### Official Review · Reviewer_FYCV · 2024-10-16

**Soundness:** 3
**Presentation:** 3
**Contribution:** 2
**Rating:** 6
**Confidence:** 3

**Summary:**

This paper reframes meta-learning as a sequence modeling problem, allowing the transformer encoder to learn task context from support images and use it to predict query images. The core is a mechanism for generating diverse tasks using a combination of data augmentations and a mixing strategy. Experimental results demonstrate the superiority over existing unsupervised meta-learning baselines, highlighting the efficacy of the model in leveraging generalization over memorization.

**Strengths:**

* The authors propose a novel task creation mechanism that generates diverse few-shot tasks from unlabeled datasets through a combination of data augmentations and a mixing strategy. This seems reasonable.

* They utilize the transformer’s in-context learning capability, eliminating the need for fine-tuning to specific test domains, which is commendable.

* The experiments are thorough, and the exploration shows that the model achieved generalization.

**Weaknesses:**

* The comparisons in the experiments may not be entirely fair. For instance, differences in model architecture between methods, and whether additional feature extractors or class encoders were used, should be noted in a table to enable clearer comparisons. To my knowledge, this seems somewhat unfair to the in-domain baselines. It would be appreciated if a table showing the parameter count and computational overhead for each method were included.

* In comparison to CAML, another transformer-based in-context learning method, or SSL methods, CAMeLU doesn’t perform as well.

**Questions:**

* What are the training costs for CAMeLU? When constructing tasks from unlabeled datasets, especially when using a mixup strategy, do the hyperparameters need to be carefully tuned?

* The comparison with CAML in Figure 3. Why doesn't CAML show an improvement in relative validation accuracy during training? What could be the reason for this?

---

> ### Author Response · Authors · 2024-11-20
> **Response to Reviewer FYCV (1/2)**
>
> We thank the reviewer for the valuable feedback. We have addressed the comments as follows:
>
> W1: As clarified in Section 4.1, using the same model architecture and scenario (in-domain vs. cross-domain) for all the proposed approaches was not feasible due to limitations of previous work. So, we follow the model architectures and the presentation proposed in PsCo and we divide the method into in-domain and cross-domain. The in-domain scenario used for evaluation in previous work is an easier setting that should favour those methods compared to what proposed in this paper, as they can extract the knowledge of the evaluation domain directly from the training phase. Indeed, initial studies in the UML literature, were not designed to be applied to the challenging cross-domain scenario, and they could not be run with very large architectures: for example, baselines like CACTUs and UMTRA are MAML-based approaches, and these known to be subject to performance degradation when a larger feature extractor is used, i.e. the model performance does not scale (and it even worsen) with the increase of the model parameters [3]. For Meta-GMVAE, instead, we replaced their initial feature extractor (Conv5) with ResNet-50 pre-trained with SimCLR on imagenet-1k to provide a fairer comparison: this, indeed, leads to increased performance than what was declared in the original paper. For all the other approaches and baselines (PsCo, BECLR, SimCLR, SwAV) we used ResNet-50 as feature extractor. Finally, we also ablate the feature extractor used in CAMeLU in Appendix A.4.\
> To further accomplish the reviewer’s request, we also included Table 11 in Appendix A.7 showing the computational overhead and the time required for running our method, compared to the latest UML approach (PsCo). The results clearly show how our model is faster both during the training and test phases and less demanding on the computational resources, as we do not require a large batch size, nor a queue that maintains the information of previous tasks in memory.
>
> W2: CAML is a fully supervised method and it was used in Table 1 as an upper bound (oracle) baseline considering that it fully utilizes labeled data during training. Nevertheless, it’s interesting to notice that CAMeLU outperforms CAML on datasets that differ significantly from the training set (e.g., CUB, Aircraft, Meta-iNat). This is primarily due to our task creation mechanism that enhances the model’s generalization ability by increasing variability across tasks. Additionally, as shown in Section 4.5, CAML struggles to generalize when the number of classes is limited, as the low variability in the training tasks restricts its ability to generalize effectively  [1,2].\
> Regarding SSL approaches, their purpose is different from UML methods. UML are trained to quickly solve the same tasks on previously unseen classes, focusing on learning a learning strategy, while SSL focus on learning robust and transferable representations for a feature extractor, so that it can adapt to different downstream tasks, provided some fine tuning steps. This results in lower performance for SSL methods when evaluated on datasets that significantly differ from the training data, as they may struggle to transfer learned features to such diverse data. Notably, SSL methods require an adaptation phase during evaluation, whereas CAMeLU performs without any fine-tuning and it predicts the query data in just 57 ms per task. Finally, CAMeLU is easily accessible for training on most of the current consumer device GPUs (we used an Nvidia 3070 Ti Laptop GPU with 8 GB of VRAM), due to the fact that it does not require a large batch size (SimCLR) or to maintain a queue of previous samples (SwAV).

---

> > ### Author Response · Authors · 2024-11-20
> > **Response to Reviewer FYCV (2/2)**
> >
> > Q1: We have now included a table (Table 11 in A.7) to clarify the time required for the task construction mechanism, the training time, and the inference time of CAMeLU compared to PsCo. We also reported the memory usage during training and inference for both approaches. As discussed, the time required for the task construction strategy is significantly low compared to PsCo. Furthermore, as shown in Table 10 (Appendix A.6), the model is not very sensitive to the choice of $\alpha$ and $\beta$ if the values guarantee the sampling of a sufficiently small $\lambda$ to ensure the incorporation of sufficient information from  $\tilde{x}_{n,j}$ into $x_j^{(qr)}$ (see results for $\alpha$, $\beta = 1$ and $\alpha=2$, $\beta=5$ in Table 10 ).
> >
> > Q2: This is an interesting insight from our paper and we hypothesize it is due to the fact that the in-context learner requires a large number of classes for generalizing, as also discussed in [1,2]. Indeed, by increasing the variability between tasks, the model can better exploit this variability to extract more information from the data, and hence solve a more complex generalization problem. We have added some details about this rationale in appendix A.5. In the case of CAML, training with only the 64 classes of miniImagenet, is not enough to acquire enough information for effective generalization. However, CAMeLU is not affected by this, as the proposed task construction strategy randomly assigns classes to instances, introducing greater variability within tasks. This enables CAMeLU to better handle more complex generalization tasks, even when a small-scale dataset is used during training.
> >
> > References:
> >
> > [1] Chan, S., Santoro, A., Lampinen, A., Wang, J., Singh, A., Richemond, P., McClelland, J., & Hill, F. (2022). Data Distributional Properties Drive Emergent In-Context Learning in Transformers. In Advances in Neural Information Processing Systems (pp. 18878–18891). Curran Associates, Inc.\
> > [2] Singh, A., Chan, S., Moskovitz, T., Grant, E., Saxe, A., & Hill, F. (2023). The Transient Nature of Emergent In-Context Learning in Transformers. In Advances in Neural Information Processing Systems (pp. 27801–27819). Curran Associates, Inc. \
> > [3] Chen, W.Y., Liu, Y.C., Kira, Z., Wang, Y.C., & Huang, J.B. (2019). A Closer Look at Few-shot Classification. In International Conference on Learning Representations.

---

> > > ### Comment · Reviewer_FYCV · 2024-11-21
> > >
> > > Thank you for addressing my concerns in your response. I am pleased with the clarification and would like to increase my score from 5 to 6.

---

### Official Review · Reviewer_j5he · 2024-11-02

**Soundness:** 2
**Presentation:** 3
**Contribution:** 3
**Rating:** 6
**Confidence:** 5

**Summary:**

This paper proposes CAMeLU, a novel approach to unsupervised meta-learning (UML) that utilizes transformer architectures' in-context learning capabilities to extract contextual information from support samples and perform predictions on query data as a sequence modeling task.

**Strengths:**

1.	Overcomes the limitation in the UML field of relying solely on simple data augmentation for constructing training tasks, by proposing a novel sequence modeling-based task construction approach.

2.	Effectively leverages the advantages of in-context learning in LLM.

3.	Provides extensive experiments and a thorough hyperparameter tuning process, which comprehensively demonstrate the advantages of the proposed algorithm.

**Weaknesses:**

1.	Lacks an explanation of the benefits of combining transformer architecture with the proposed data construction method, and does not clarify whether other architectures could also be adapted. In short, there is insufficient discussion of the model's generalizability.

2.	Lacks detailed descriptions of the fixed feature extractor f and the learned class encoder g.

**Questions:**

1.	During the construction of the query set using mixup, does the correlation between the sampled image and the combined augmented image affect the model's performance and stability? For instance, do cases where they belong to the same class or are completely unrelated impact the outcome?

2.	Under time constraints, does the computational cost of this construction method lead to a significant performance improvement in the model?

3.	How sensitive is the model's performance to the parameters used in the query set construction method?

---

> ### Author Response · Authors · 2024-11-20
> **Response to Reviewer j5he (1/2)**
>
> We thank the reviewer for the valuable feedback and we have now addressed the reviewer’s concerns as follows:
>
> W1: We have included more details about the importance of combining the proposed task creation mechanism with an in-context learner in Section 3. Furthermore, we have also added more experiments to verify the applicability of our proposed task creation strategy with other architectures (see Appendix A.5).\
> The combination of a transformer architecture with the proposed task creation mechanism is central to our approach, as it leverages the in-context learning capabilities of transformers to address the challenges of unsupervised meta-learning. Transformers excel at modelling dependencies and capturing relationships between support and query samples, which is especially beneficial in few-shot learning scenarios. Our task creation mechanism complements this by constructing diverse and challenging pseudo-tasks, effectively preparing the model for the complexities of test-time tasks. However, the proposed mechanism is not restricted to our model architecture, but it can be easily combined with other meta-learning methods. To demonstrate this, we conducted additional experiments by applying the proposed task creation on top of MAML. As shown in Table 8 in Appendix A.5 and discussed in the text, this approach outperforms UMTRA and CACTUs-MAML, where the only difference among the three is the task construction mechanism, further demonstrating that our task creation method is versatile and adaptable to other architectures. The results also reveal that without the transformer-based ICL capabilities, the performance falls significantly short of CAMeLU, highlighting the critical role of combining these two components.
> The importance of task creation for cross-domain generalization is further underscored in Table 7 (Appendix A.5), which shows an ablation study by combining different techniques (e.g., k-means clustering and only data augmentations) into CAMeLU.
> Moreover, Figure 4 in Appendix A.2 provides evidence of how the transformer encoder leverages the entire task context to predict the query label, enhancing the learned representation and ultimately improving the accuracy.
>
> W2: Details of the feature extractor are mentioned in Section 4.2 (specifically lines 359-360), and more specifications are detailed in A.1. Furthermore a complete ablation study is presented in Section A.4. For the class encoder $g_{\phi}$ it is as simple as a linear layer that maps one-hot encoded vectors of labels in the range [0, N-1] to a dimensionality of 256, as stated in A.1. To clarify this further, we have added an additional explanation in Section 4.2, as suggested. Furthermore, we have shared our code alongside the manuscript, which contains all the details of our model and ensures the reproducibility of our experiments.
>
> Q1: This is an interesting concern and we have found that having some correlation between the sampled and the combined augmented image, for some tasks, might even be beneficial for the model's robustness. As Equation at line 199 proves, we can treat each sample as an instance class as long as we use a large-scale training dataset with a large number of classes (964, in our case). If two images belonging to the same original class are sampled within the same episode, this will result in a more challenging task for the model to solve. On the other hand, other tasks might be very easy, based on the nature of the N sampled tasks. This variability in task difficulties will make the generalization problem harder and the in-context learner more robust, as it will learn to solve a variety of different tasks with different levels of complexity, which better mimic the few-shot target tasks observed at test time. This claim is supported by the findings in [1,2] and from the results in Figure 5 and Appendix A.5, which demonstrate that our choice of using a task creation mechanism inspired by mixup makes the tasks more difficult to solve, compared to apply simple data augmentations, hence enhancing the model robustness. We have clarified this in Appendix A.5.
>
> Q2: To clarify the computational cost of our task construction strategy, we have added a comparison with PsCo, the current SOTA method in UML, in Table 11 (Appendix A.7). The results show that the task construction method used by CAMeLU is 15 times faster than PsCo, with a task construction time of only 1376 ms. In comparison to other UML baselines, the task construction time of CAMeLU is comparable, or even lower: UMTRA requires 104 ms, while CACTUs requires 104361 ms. These results highlight that the performance improvements achieved by CAMeLU do not come at the cost of significantly increased computational time for generating pseudo-tasks, ensuring that our method remains efficient and scalable.

---

> > ### Author Response · Authors · 2024-11-20
> > **Response to Reviewer j5he (2/2)**
> >
> > Q3: The hyperparameters involved in the query data creation are detailed and ablated in Table 10 in Appendix A.6. Here, we run the experiments with different values of $\alpha$ and $\beta$, which are the parameters of the $Beta$ distribution from which the $\lambda$ value used for our mixup strategy is sampled from(see equation 1 for clarity). The results show that the model is not very sensitive to the choice of $\alpha$ and $\beta$ if the values guarantee the sampling of a sufficiently small $\lambda$ that ensure the incorporation of sufficient information from  $\tilde{x}_{n,j}$ into $x_j^{(qr)}$ (see results for $\alpha, \beta = 1$ and $\alpha=2, beta=5$ in Table 10).
> >
> > References:
> >
> > [1] Chan, S., Santoro, A., Lampinen, A., Wang, J., Singh, A., Richemond, P., McClelland, J., & Hill, F. (2022). Data Distributional Properties Drive Emergent In-Context Learning in Transformers. In Advances in Neural Information Processing Systems (pp. 18878–18891). Curran Associates, Inc.\
> > [2] Singh, A., Chan, S., Moskovitz, T., Grant, E., Saxe, A., & Hill, F. (2023). The Transient Nature of Emergent In-Context Learning in Transformers. In Advances in Neural Information Processing Systems (pp. 27801–27819). Curran Associates, Inc.\

---

> ### Author Response · Authors · 2024-11-25
> **Follow up**
>
> We sincerely thank the reviewer for the time and effort you have dedicated to reviewing our paper and for your valuable feedback. As the deadline for providing answers and clarifications is approaching, we kindly wanted to follow up to check if you have had the chance to review our responses and the discussion with the other reviewers. If there are any additional concerns or questions, we are available to address them promptly.
> Thank you once again for your contributions and for helping us refine our manuscript.

---

> > ### Author Response · Authors · 2024-11-29
> > **Final follow up**
> >
> > Once more, we thank the reviewer for the initial feedback provided during the first rebuttal phase. As the deadline for the rebuttal process is soon approaching we would like to confirm whether we have fully addressed the reviewer's concerns. Our effort and engagement with the other reviewers have resulted in an increase in their overall score. We are committed to ensure all reviewers are fully satisfied with our work and its impact. As such, we are available to provide further clarifications and revisions if there are more questions or concerns.

---

### Official Review · Reviewer_HhvW · 2024-11-03

**Soundness:** 3
**Presentation:** 2
**Contribution:** 3
**Rating:** 6
**Confidence:** 2

**Summary:**

The paper develops CAMeLU for unsupervised meta-learning setting, where the goal is to leverage the in-context learning capabilities of transformers to create transferable feature representations from unlabeled data for new tasks with small numbers of labeled data. By modeling meta-learning as a sequence prediction problem, each observed task is modeled as a sequence of support instances and an unknown query image. The transformer encoder subnetwork then identifies task-specific patterns from support images for that task which enhances accuracy of predictions on query images by making the model specialized for the input task. Data augmentation and task mixing are also used to generate diverse training tasks that challenge the model to improve generalization on new tasks. Experimental results on a benchmark formed from five datasets are offered to demonstrate that CAMeLU is effective for unsupervised meta-learning and leads to a performance close to supervised and self-supervised meta-earning methods.

**Strengths:**

1. The paper is relatively straightforward to read and follow.

2. The performance boost compared to the baselines is significant on all benchmarks.

3. The code is provided and although I did not run it but it looks well documented and easy to run which will make reproducing the results straightforward.

**Weaknesses:**

1. The model may become very large due to the added transformer model which computationally may become expensive and challenging to handle compared to methods are using only CNN architectures.

2. Comparison is limited to only a few methods, despite the fact meta learning have a rich literature.

3. Analytic experiments are limited and do provide much insight about the proposed method. For example, what are the circumstance and criteria under which the proposed method is more effective? What are the limitations? What is the reason behind the improved performance compared to the baselines?

**Questions:**

1. I was wondering if mixup is used during model training, would we still get a reasonable performance in a meta learning setting?


2. What is the specific justification for the design that was used for the transformer model that is used for experiments? What is the computational overhead that it adds?

3. Metalearning is overcrowded field with so many existing works. What was the reasoning to select methods that were used for comparison? It looks some of these methods are not the most recent methods which likely will have better performance. It is necessary to include 8-10 methods for comparison with majority of them to be most recent works that achieve the highest performances to demonstrate that the proposed method is competitive.

---

> ### Author Response · Authors · 2024-11-20
> **Response to Reviewer HhvW (1/2)**
>
> We thank the reviewer for the useful comments. We have answered to the questions as follows:
>
> W1: While we acknowledge the reviewer’s concern regarding the increased computational expense and model size associated with transformer models, we have demonstrated that CAMeLU achieves state-of-the-art performance with a relatively compact architecture. Specifically, it uses just 8 encoder layers, each with 8 attention heads, which is significantly smaller than the models typically used in the literature (e.g., see Table 1 in [1]). This allows the model to fit on a mid-range consumer device with a single laptop GPU. Details about the architecture and memory requirements can be found in Appendix A.1.
> To further clarify the computational cost, we have included Table 11 in Appendix A.7 that outlines the time and memory required for both training and inference. Our results show that CAMeLU is much faster (by an order of magnitude) compared to the state-of-the-art UML approaches and requires only one-sixth of the GPU memory. Additionally, in Appendix A.7 we have analysed the computational complexity of our approach, highlighting that good performance can be achieved with only one support sample per class K and one query sample Q. This results in a complexity $O(N^2 * d)$. For few-shot learning scenarios, while $N$ is usually small, $d$ can become quite large, depending on the original image size and the chosen feature extractor. Nonetheless, the training time for CAMeLU is still significantly lower than other methods and the model’s scalability (as discussed in Appendix A.7) makes it particularly appealing for researchers, as it can be easily trained and evaluated on small, portable devices.
>
> W2: For comparison evaluation, we focused on methods that are mostly relevant to the specific problem addressed by our work, which is unsupervised meta-learning. We compare our approach with five existing unsupervised meta-learning methods (with PsCo the current state-of-the-art), an unsupervised few-shot learning method which is the state-of-the-art on cross-domain scenarios (BECLR), two SSL methods (SimCLR and SwAV) and the supervised counterpart of CAMeLU, i.e., CAML. We didn’t consider other supervised meta-learning methods, as they are out of the scope of our paper, which instead focuses on unsupervised meta-learning. These baselines provide a comprehensive evaluation of CAMeLU’s performance and provide interesting insight on its generalization capabilities. Additionally, the compared baselines are in line with the previous literature, particularly with the most recent SOTA unsupervised meta-learning approach, i.e., PsCo.
>
> W3: We believe the experiments presented in the main text and appendix provide a thorough evaluation of CAMeLU’s performance, particularly under cross-domain scenarios. These experiments aim to highlight the method's strengths, insights, and limitations. Specifically, we have conducted the following:
>  - Cross-domain evaluation and comparison with previous methods (4.3, 4.6);
>  - Generalization capabilities of CAMeLU and learning curves analyses  (4.4, A.4);
>  - Insights into the CAMeLU’s abilities to learn from the context task and exploit the ICL ability of transformers (A.2, A.8);
>  - Scalability of CAMeLU (4.5, A.3, A.7);
>  - Analyses of the proposed task creation strategy (A.5, A.6);
>
> During this rebuttal period, we have also added an experiment to prove the versatility of our task creation mechanism which can be applied also with existing baselines (e.g., UMTRA) in Table 8 in A.5. We also added more details about the computational overhead and time required for training and testing CAMeLU (Table 11 in Appendix A.7). This, together with the Limitations section in Appendix A.9 and the commit of releasing our code (already available for reviewers as part of the supplementary material) and pre-trained models upon acceptance of the paper, ensure the trustworthiness of our method.
> We are open to add more experiments, as far as they are relevant and they can be written, executed, evaluated and eventually discussed during the remaining period of the rebuttal.

---

> ### Author Response · Authors · 2024-11-20
> **Response to Reviewer HhvW (2/2)**
>
> Q1: To address your question, we have provided two possible interpretations, as we are not sure to have fully understood the intent of your inquiry. Please, feel free to clarify if we have missed the core of your concern.
>
>  - (a) In our approach we used a technique inspired by mixup for task construction as we described in Section 3.1. The goal is to generate training tasks with enhanced variability between support and query images compared to using the same set of augmentations (e.g., as in UMTRA). This is done in the attempt to better resemble the difficulty of the test tasks, where query and support samples are different images belonging to the same classes. At test time, indeed, standard few-shot tasks are available and they do not need to be constructed. In contrast, conventional mixup aims to create new virtual examples and labels that extend the distribution, thus resulting in a less confident (smoother, less peaked) decision boundary during the training phase and avoiding overfitting. The distinction between our proposed technique and conventional mixup is explained in Section 3.1.
>
>  - (b) If the question pertains to applying standard mixup within conventional meta-learning algorithms to mitigate overfitting, we believe there is no inherent limitation preventing its application. Such an approach could potentially increase performance by enhancing the variability of the training data and generate a smoother decision boundary during the training phase.
>
> Q2: For the chosen architecture we took inspiration from the ViT-Base architecture proposed in [1], we removed the initial patch linear projection layer and we replaced the final MLP with a single linear layer. We then reduced the dimensionality of the transformer architecture to overcome the hardware limitations of our laptop GPU.
> To better clarify the computational overhead and time required for training and testing CAMeLU we have added Table 11 in Appendix A.7 detailing the computational and time complexity for training and inference with CAMeLU, compared to PsCo. Our results show that CAMeLU is much faster (by an order of magnitude) compared to the state-of-the-art UML approaches and requires only one-sixth of the GPU memory. Additionally, we have also analysed the computational complexity for the proposed model in Appendix A.7.
>
> Q3: Please see W2.
>
> References:
>
> [1] Dosovitskiy, A. (2020). An image is worth 16x16 words: Transformers for image recognition at scale. arXiv preprint arXiv:2010.11929.

---

> > ### Comment · Reviewer_HhvW · 2024-11-21
> > **Post-Rebuttal Rating**
> >
> > I checked the rest of reviews and the responses you provided. Thank you for the through responses. I increased my final rating accordingly.

---

### Official Review · Reviewer_Lqp1 · 2024-11-03

**Soundness:** 4
**Presentation:** 3
**Contribution:** 3
**Rating:** 6
**Confidence:** 5

**Summary:**

This work aims to tackle few-shot learning (FSL) by learning from unsupervised dataset. It shares a similar idea with CAML[1] in model design by taking FSL as a sequence modeling problem. The instance embedding $f_{\psi}$ and the class encoding $g_{\phi}$ of
 a support image are contacted to form a "token" in the context. of the sequence, and a transformer encoder learns from the context to generated output feature for the query. The output feature finally is projected by a top linear layer predict the class of the query.

To learn from the given unsupervised dataset, this work proposes a novel task creation mechanism to generate pseudo-tasks. In each run of the task creation mechanism, $N$ samples are randomly selected from the unsupervised dataset and data augmentation strategies are applied to each sample to generate $K$ instances (which are utilized as the support examples), simulating the $N$-way $K$-shot setting. Query samples in the pseudo-task are generated by mixup the selected sample (and its augmented version) with randomly selected samples in the "background" dataset.

The pseudo-task creation provides a way to perform meta-learning in the pre-training stage when no labeled dataset is available, which is interesting and instructive.


[1] Fifty, C., Duan, D., Junkins, R. G., Amid, E., Leskovec, J., Re, C., & Thrun, S. Context-Aware Meta-Learning. In The Twelfth International Conference on Learning Representations.

**Strengths:**

1) The work proposes a novel pseudo-task creation mechanism to generate FSL tasks out of unsupervised dataset, providing a new (maybe efficient) way to learn from unlabeled data.
2) Experiments with CAML show that CAMeLU outperforms other unsupervised methods to achieve promising results.

**Weaknesses:**

However, I have the following concerns:
1) The model in this work is the same as CAML, which shows limited novelty.
2) Though the task formulation is interesting, only CAML and its successor (e.g. CAMeLU) are fairly trained in the setting. Other unsupervised method cannot leverage the supervised pseudo-tasks as supervised methods do. Essentially, CAMeLU is a supervised method. The pseudo-task generation provides a way to bridge the unsupervised scenario and the supervised scenario.
3) Instead of providing a specific instance (e.g. CAML) to illustrate how the proposed task generation facilitates FSL from unsupervised dataset. I prefer to see more evidences that showcase the pseudo-task creation is a valid and efficient mechanism that makes supervised method like ProtoNet, MAML, etc, feasible in unsupervised scenario. This, I think, is the core contribution of this work.
4) In experiment, building a new subset of Imagenet-1k (i.e. ImageNet-964) is not necessary since the method is so-called unsupervised. Note that, CLIP and its variants would cause information leakage if not carefully considered, because the texts may explicitly work as anchors to categorize instances belong to the same class together.
5) I donot think it is proper to put experiments on CIFAR-FS into the cross-domain setting, because CIFAR-FS contains natural scene color images as in the miniImagenet. Conventionally, cross-domain setting refers to either transferring from coarse-grained setting to fine-grained setting as in FSL, or transferring from natural scene color image setting to painting image setting where instances from two domains show quite different appearances.
6) Typos: e.g. In A.1 Line 793, the iamge embedding's output size decreases to 768, resulting ....

**Questions:**

One more question:
As the class of instances in FSL are pseudo-labels randomly picked in range $0-N$, how to explain the top linear lay for classification that project the output embedding of the query into the final category distribution? An instance could be of class 0 in task $T_i$ but of class 1 in task $T_{i+1}$, how can the in-context classifier (i.e. the top linear lay) predict its class without finetuning upon the support set?

---

> ### Author Response · Authors · 2024-11-20
> **Response to Reviewer Lqp1 (1/2)**
>
> We thank the reviewer for the useful feedback. We address your comments below.
>
> W1: While the proposed approach shares some similarities with CAML, there are several in-context learning (ICL) methods which use similar transformer-based architectures [1,2,3,4]. Specifically, our approach builds on the standard practice in ICL literature of employing a transformer-based model paired with a simple linear layer to learn embeddings of input labels (or tokens). Furthermore, we used off-the-shelf embedding features of known classes (represented by the fixed feature extractor) allowing the in-context learner to work on pre-computed features rather than learn embeddings from scratch. This approach is frequently used in ICL models that deal with complex input data [1,2,3,4].
> The only characteristic shared exclusively with CAML (and not with other ICL methods) is the non-causal nature of the task sequence, which is a typical assumption in meta-learning. We had already acknowledged this similarity throughout the paper (lines 344, 345) and we had performed several experiments to compare the two methods (Table 1; the whole Section 4.5; Table 5). We have further clarified similarities and differences in Section 2 of the updated manuscript. Furthermore, our work significantly differs from previous ICL methods as we incorporate a novel unsupervised task creation mechanism that automatically generates tasks without labeled data. This contribution bridges the gap between the supervised and unsupervised setting as intended and, combined with the ICL capability of transformer architectures, enables the model to leverage task context for accurate predictions in the unsupervised scenario.
>
> W2: We agree that pseudo-task generation acts as a bridge between unsupervised and supervised scenarios, and this bridging is indeed the central goal of unsupervised meta-learning methods, as outlined in the related work section and supported by prior survey papers [5,6]. One of the main challenges in meta-learning is constructing tasks from an unlabeled dataset during the meta-training phase in a way that they closely resemble those encountered during the supervised meta-test, where original images (not augmented) and labels are available. This is necessary to ensure the model generalizes during evaluation on few-shot tasks starting from a different, unsupervised training setting. Previous studies have struggled to address this issue, and they have searched for several methodologies to construct tasks for unsupervised meta-training.
> To demonstrate the broader applicability of our proposed task creation mechanism, we conducted additional experiments where we replaced the task creation mechanism of MAML-based unsupervised meta-learning baselines (i.e. UMTRA, CACTUs-MAML) with our method. As shown in Table 8 in Appendix A.5, our method improves the performance of these baselines, demonstrating its superiority over previous task creation methods. It is worth noting, however, that CACTUs-MAML achieves higher performance on *mini*ImageNet primarily due to data leakage from its feature extractor, which was pre-trained on ImageNet-1k. This results in an unfair advantage, as it introduces information about the test data into the training phase. This performance gap shrinks when the test distribution deviates from the training data (e.g., CIFAR-fs) and eventually disappears in all other test sets that share low correlation with ImageNet-1k, thus confirming our hypothesis.
> Nonetheless, even with this improvement, the performance remains notably lower than CAMeLU, emphasizing the unique advantage of CAMeLU in leveraging the in-context learning capabilities of transformer architectures for accurate predictions, particularly in cross-domain scenarios.
>
> W3: Please see the previous response.
>
> W4: In meta-learning, the goal is to enable models to quickly adapt to new, unseen classes at test time [5,6]. Using ImageNet-1k would violate this fundamental assumption that the training and test classes must be disjoint, as *mini*ImageNet is a subset of ImageNet-1k. This overlap would cause the model to encounter instances from test classes during training, leading to potential memorization rather than true generalization. Even in an unsupervised setting, this would expose the model to learn some class features, despite the inability to link them to a precise class, thus compromising the integrity of the evaluation.
> To address this issue and ensure a fair evaluation, we introduced ImageNet-964 which is simply a subset of ImageNet-1k, where classes belonging to the validation and test splits of *mini*ImageNet - used during model evaluation - are removed. This step prevents any leakage of information, maintaining the integrity of the evaluation process.
> Regarding the experiments with CLIP, we recognize that preventing data leakage is challenging as the dataset used for training CLIP is not publicly available and we have now added a clarification in Appendix A.4.

---

> ### Author Response · Authors · 2024-11-20
> **Response to Reviewer Lqp1 (2/2)**
>
> W5: While we acknowledge that CIFAR-fs and *mini*ImageNet both consist of natural scene color images, we adhere to the definition of "cross-domain" established in prior literature on cross-domain meta-learning and few-shot learning (e.g., PsCo and BECLR), which defines this scenario as one where training and testing datasets are different. While this definition does not impose stricter requirements on the degree of visual or semantic difference between the domains, it still represents a challenging meta-learning scenario.
> We recognize that CIFAR-fs and *mini*ImageNet share certain similarities (as mentioned e.g., in Sections 4.1, 4.4, 4.6), but CIFAR-fs has a smaller resolution (32x32) compared to *mini*ImageNet. This difference introduces an additional challenge for cross-domain generalization. Furthermore, our choice to include CIFAR-fs stems from its extensive use as a benchmark in meta-learning literature, enabling fair and consistent comparisons with existing baselines, as well as the fact that it shares the same number of classes, images per class, and split ratios of *mini*ImageNet.
> Additionally we also evaluated CAMeLU on datasets with significant domain differences, such as CUB, Aircraft, and Meta-iNat. These datasets represent more diverse cross-domain scenarios, and the results reported in the manuscript highlight CAMeLU's ability to generalize effectively even in such challenging conditions.
>
> W6: Thank you. We have now improved the sentence and checked the whole text again for other typos. All the changes are highlighted with a blue color.
>
> Q1: The top linear layer in our framework serves as a mapping mechanism to convert the output embedding of the query (produced by the transformer) into a probability distribution over the N categories. It is important to note that this linear layer is not designed to memorize specific class identities across tasks but rather to generalize across the representations learned by the in-context learner (the transformer).
> The in-context learner operates by embedding both the support set and the query within a shared latent space. During training, the model learns to associate input-output pairs (i.e., the task context) such that the query embedding aligns closely with the embeddings of the relevant support class, regardless of the specific class labels assigned in a given task. This alignment is achieved through the transformer’s ability to encode task relationships and effectively condition on the task context. Thus, while the class identities are task-specific and randomized (e.g., the same instance is assigned class 0 in task A and class 1 in task B) , the model learns to predict the relative relationships between the query and the support set. The linear layer simply projects the query embedding into the task-specific class probabilities, leveraging the relationships encoded by the transformer.
> This behavior is similar to what has been observed in models like BOIL [7] and CAML [4], where the representation space learned by the model ensures that the classifier head (linear layer) can remain fixed and still deliver accurate predictions based on the task-specific embedding relationships. By not requiring fine-tuning of the classifier head, the model maintains its efficiency and adaptability across diverse tasks.
>
> References:
>
> [1] Louis Kirsch, James Harrison, Jascha Sohl-Dickstein, & Luke Metz. (2024). General-Purpose In-Context Learning by Meta-Learning Transformers.
>
>
> [2] Chan, S., Santoro, A., Lampinen, A., Wang, J., Singh, A., Richemond, P., McClelland, J., & Hill, F. (2022). Data Distributional Properties Drive Emergent In-Context Learning in Transformers. In Advances in Neural Information Processing Systems (pp. 18878–18891). Curran Associates, Inc.
>
>
> [3] Singh, A., Chan, S., Moskovitz, T., Grant, E., Saxe, A., & Hill, F. (2023). The Transient Nature of Emergent In-Context Learning in Transformers. In Advances in Neural Information Processing Systems (pp. 27801–27819). Curran Associates, Inc.
>
>
> [4] Christopher Fifty, Dennis Duan, Ronald Guenther Junkins, Ehsan Amid, Jure Leskovec, Christopher
> Re, and Sebastian Thrun (2024). Context-aware meta-learning. In The Twelfth International Conference
> on Learning Representations.
>
>
> [5] Hospedales, T., Antoniou, A., Micaelli, P., & Storkey, A. (2021). Meta-learning in neural networks: A survey. IEEE transactions on pattern analysis and machine intelligence, 44(9), 5149-5169.
>
>
> [6] Vettoruzzo, A., Bouguelia, M. R., Vanschoren, J., Rognvaldsson, T., & Santosh, K. C. (2024). Advances and challenges in meta-learning: A technical review. IEEE Transactions on Pattern Analysis and Machine Intelligence.
>
>
> [7] Oh, J., Yoo, H., Kim, C., & Yun, S. Y. (2021) BOIL: Towards Representation Change for Few-shot Learning. In International Conference on Learning Representations.

---

> ### Author Response · Authors · 2024-11-25
> **Follow up**
>
> We sincerely thank the reviewer for the time and effort you have dedicated to reviewing our paper and for your valuable feedback. As the deadline for providing answers and clarifications is approaching, we kindly wanted to follow up to check if you have had the chance to review our responses and the discussion with the other reviewers. If there are any additional concerns or questions, we are available to address them promptly.
> Thank you once again for your contributions and for helping us refine our manuscript.

---

> ### Comment · Reviewer_Lqp1 · 2024-11-27
>
> I read the response. I thoroughly understand how CAMeLU functions and how it benefits FSL from unsupervised data. However, as said, the core contribution is the pseudo task generation but not the model ( which shares too many similarities with the supervised CAML). The pseudo task generation should be a plug-and-play component that could be combined with the many existing supervised FSL method. In essence, CAMeLU is a supervised FSL method not just as suggested by the title.
>
> All in all, I agree to increase score from 3 to 5, though I am not fully convinced.
>
> I notice that [Chris Fifty](https://openreview.net/profile?id=~Christopher_Fifty2) who is the author of CAML came and discussed with us reviewers. Many thanks for that.

---

> ### Author Response · Authors · 2024-11-27
> **Response to Reviewer Lqp1 - Round 2**
>
> We sincerely thank the reviewer for engaging in the discussion and taking the time to further discuss with us the contribution of CAMeLU. \
> We completely agree that our pseudo-task generation approach should be a plug-and-play component that could be combined with the many existing supervised few-shot learning methods. In fact, as suggested in the previous comment, we have conducted additional experiments (Table 8 in Appendix A.5 and in the general response) to demonstrate the possibility of applying our task creation mechanism to other methods, for instance MAML-based approaches. Our results demonstrate that the task creation mechanism on its own is not enough to guarantee SOTA performance. Particularly in the challenging cross-domain scenario, the combination of an in-context learner with our pseudo task generation approach proves to work much better.
> Furthermore, as defined in previous unsupervised meta-learning (UML) literature [1,2,3], the core of UML lies in the creation of training tasks from an unlabeled dataset that allows the model to learn to generalize to the supervised few-shot tasks encountered during test time. To do so, PsCo, the previous SOTA in the UML literature, borrowed concepts from the SSL literature (the queue, as in MoCo [4], and the teacher-student model updated by the exponential moving average as in BYOL [5]) for effective UML performance. Similarly, we took inspiration from the in-context learning (ICL) literature to further improve the performance while reducing the computational resources required for training and inference compared to PsCo (as demonstrated in Table 11 in Appendix A.7 and in the general response) thanks to the removal of the queue system and the teacher-student architecture. Finally, the incorporation of our task creation mechanism enables the use of smaller training datasets (e.g., *mini*ImageNet) to achieve the desired generalization, which is not feasible using only CAML, as shown in Figure 3 in our manuscript. Indeed, the task creation mechanism can be interpreted as a task augmentation strategy which improve the generalizability of the original CAML model even further.
>
> References:\
> [1] Vanschoren, Joaquin. "Meta-learning." Automated machine learning: methods, systems, challenges (2019): 35-61.\
> [2] Hospedales, T., Antoniou, A., Micaelli, P., & Storkey, A. (2021). Meta-learning in neural networks: A survey. IEEE transactions on pattern analysis and machine intelligence, 44(9), 5149-5169.\
> [3] Vettoruzzo, A., Bouguelia, M. R., Vanschoren, J., Rognvaldsson, T., & Santosh, K. C. (2024). Advances and challenges in meta-learning: A technical review. IEEE Transactions on Pattern Analysis and Machine Intelligence.\
> [4] He, K., Fan, H., Wu, Y., Xie, S., & Girshick, R. (2020). Momentum contrast for unsupervised visual representation learning. In Proceedings of the IEEE/CVF conference on computer vision and pattern recognition (pp. 9729–9738).\
> [5] Grill, J.B., Strub, F., Altche, F., Tallec, C., Richemond, P., Buchatskaya, E., Doersch, C., Avila Pires, B., Guo, Z., Gheshlaghi Azar, M., & others (2020). Bootstrap your own latent-a new approach to self-supervised learning. Advances in neural information processing systems, 33, 21271–21284.

---

### Author Response · Authors · 2024-11-20
**General response**

We would like to thank each reviewer for the valuable time and effort spent reviewing our manuscript.

As reviewers highlighted, we propose a novel task creation mechanism to generate tasks from unsupervised data (reviewers Lqp1, j5he, FYCV) and we reframe meta-learning in the context of in-context learning for leveraging transformer’s capabilities (reviewers j5he, FYCV), thus eliminating the need for fine-tuning to test domains. We conducted extensive experiments to demonstrate the superiority of our approach compared to previous baselines (reviewers Lqp1, HhvW, j5he, FYCV) and we made the code available for easy reproducibility.

We have carefully considered and addressed all reviewers' comments. We have updated the manuscript (changes in blue) with the following additional discussions and experiments:
  - We have further clarified the contributions of existing in-context learning approaches relevant to our study and we have highlighted the main differences with our proposed method in Section 2.
  - We have highlighted in Section 3 the importance of considering the combination of our task creation strategy together with an in-context learner to achieve superior cross-domain performance.
  - We have conducted additional experiments (Table 8 in Appendix A.5 and the relative discussion in lines 1066-1080), to demonstrate the versatility of the proposed task creation mechanism which can be easily used in combination with other meta-learning approaches to enhance the unsupervised meta-learning performance. It is worth noting that the performance of CACTUs-MAML is affected by data leakage from its feature extractor (with *mini*ImageNet), but the performance gap decreases as the data distribution deviates from the training data. Nonetheless, the performance remains significantly lower than CAMeLU, emphasizing the unique advantage of CAMeLU in leveraging the in-context learning capabilities of transformer architectures.

>**Table 8 in Appendix A.5:**
| Method             | miniImageNet 5w1s | miniImageNet 5w5s | CIFAR-fs 5w1s | CIFAR-fs 5w5s | CUB 5w1s | CUB 5w5s | Aircraft 5w1s | Aircraft 5w5s | Meta-iNat 5w1s | Meta-iNat 5w5s |
|--------------------|-------------------|-------------------|---------------|---------------|----------|----------|----------------|----------------|----------------|----------------|
| **CACTUs-MAML**    | **43.30**         | **54.21**         | **42.00**     | **56.64**     | 31.19    | 36.81    | 24.06          | 27.26          | 20.13          | 21.84          |
| **UMTRA**          | 39.93            | 50.73            | 32.93        | 46.13        | 27.06    | 36.60    | 22.40          | 31.73          | 28.96          | 37.12          |
| **Proposed + MAML**| 34.04            | 46.13            | 37.02        | 52.00        | **31.22**| **41.54**| **26.12**      | **34.34**      | **30.94**      | **42.43**      |


  - We have added more details about the computational and time complexity of CAMeLU in Appendix A.7 (Table 11), by considering the time required for the task construction mechanism, the training time, the inference time and the GPU and CPU memory consumption during training and inference of the model. These results are compared with PsCo, which is the current state-of-the-art in the unsupervised meta-learning field and demonstrate that CAMeLU is significantly faster and memory efficient compared to it.

>**Table 11 in Appendix A.7:**\
> Time complexity
| Method   | Time Task Construction (ms) | Training Time (ms/epoch) | Inference Time (ms/task) |
|----------|-----------------------------|--------------------------|--------------------------|
| **PsCo** | 20772                       | 4613656                  | 605                      |
| **CAMeLU**| 1376                       | 153000                   | 57                       |

> Memory usage
| Method   | GPU Training (MiB) | CPU Training (MiB) | GPU Inference (MiB) | CPU Inference (MiB) |
|----------|--------------------|--------------------|---------------------|---------------------|
| **PsCo** | 43904              | 20904              | 1630                | 2061                |
| **CAMeLU**| 6250              | 2588               | 3224                | 1667                |

We hope our response and revision sincerely address all the reviewers’ concerns and we are open for further discussions during the rebuttal period.

---

### Public Comment · ~Chris_Fifty1 · 2024-11-21
**Public Discussion Phase Remark**

Was doing some late-night reading of meta-learning ICLR submissions and stumbled across this submission. Very cool work & interesting. Really like A.2; wish you could have brought it up into the main text---this shows why CAMeLU works & what's happening under the hood in a clear, simple way. Just a suggestion; I think it's understandings like this that'll help steer future method development.

Will be sharing this with other meta-learning researchers.

---

> ### Author Response · Authors · 2024-11-21
> **Response to public comment**
>
> We sincerely appreciate your thoughtful feedback and interest in our work. We fully intend to integrate this analysis into the main text in the camera-ready version, as we recognize the importance of the results in Appendix A.2. These results provide valuable qualitative insights into the in-context learning process of our model and its integration into the main text will strengthen the presentation and provide additional clarity for readers exploring the mechanisms behind CAMeLU. For this version of the manuscript, we chose to keep this analysis in the appendix to allow more room for discussion during the rebuttal phase while ensuring consistency with the figure and line references provided to other reviewers.

---

### Meta-Review · Area_Chair_6o76 · 2024-12-24

**Metareview:**

This work addresses the challenge of few-shot learning (FSL) by leveraging an unsupervised dataset. It adopts a similar approach to CAML [1] in its model design, framing FSL as a sequence modeling problem. This allows the transformer encoder to learn task context from support images and utilize it to predict query images. To effectively learn from an unsupervised dataset, the authors propose a novel task creation mechanism that combines data augmentation with a mixing strategy to generate pseudo-tasks. Experimental results demonstrate significant improvements over existing unsupervised meta-learning baselines, underscoring the model's ability to generalize effectively rather than rely on memorization.

All reviewers agreed that the pseudo-task creation mechanism is both novel and promising, with the empirical results showing extensive and significant improvements that validate its effectiveness.

Given the potential impact of this work on the meta-learning and few-shot learning communities, I recommend its acceptance.

**Additional Comments On Reviewer Discussion:**

During the discussion period, all reviewers actively engaged with the authors and expressed satisfaction with the responses provided. All concerns have been addressed, and the authors are encouraged to incorporate the reviewers' suggestions, particularly by moving some of the newly added appendix results into the main text, as this is considered highly important.

---

### Decision · Program_Chairs · 2025-01-22

Accept (Poster)